# DNABERT-S: Pioneering Species Differentiation with Species-Aware DNA Embeddings

## Abstract

We introduce DNABERT-S, a tailored genome model that develops species-aware embeddings to naturally cluster and segregate DNA sequences of different species in the embedding space. Differentiating species from genomic sequences (i.e., DNA and RNA) is vital yet challenging, since many real-world species remain uncharacterized, lacking known genomes for reference. Embedding-based methods are therefore used to differentiate species in an unsupervised manner. DNABERT-S builds upon a pre-trained genome foundation model named DNABERT-2. To encourage effective embeddings to error-prone long-read DNA sequences, we introduce Manifold Instance Mixup (MI-Mix), a contrastive objective that mixes the hidden representations of DNA sequences at randomly selected layers and trains the model to recognize and differentiate these mixed proportions at the output layer. We further enhance it with the proposed Curriculum Contrastive Learning ($C^2LR$) strategy. Empirical results on 23 diverse datasets show DNABERT-S's effectiveness, especially in realistic label-scarce scenarios. For example, it identifies twice more species from a mixture of unlabeled genomic sequences, doubles the Adjusted Rand Index (ARI) in species clustering, and outperforms the top baseline's performance in 10-shot species classification with just a 2-shot training. [1]

## 1 Introduction

Accurate differentiation of species from genomic sequences is a critical task in biology and ecology, supporting efforts in biodiversity conservation, epidemiology, understanding evolutionary processes, and exploring the roles of microbiomes in health and disease. Traditional methods for species identification rely heavily on well-characterized reference genomes for comparative analysis. Thus, they are limited due to the vast and largely unexplored genetic diversity present in natural environments.

A prime example is metagenomics binning. Metagenomics binning (Kang et al., 2015; 2019; Nissen et al., 2021; Meyer et al., 2022; Lamurias et al., 2023) is a pivotal process in microbiome research, aiming to group DNA sequences by species from complex mixtures containing DNA from potentially thousands of distinct, often uncharacterized species. In this context, effective DNA embeddings that can accurately segregate and cluster DNA sequences are more suitable than the methods that rely on known reference genomes for comparison and alignment.

Despite the critical role of DNA embeddings in various scenarios, there is a notable deficiency in the development of effective methods. Current approaches to achieving DNA embeddings include: 1) Descriptive textual features (Kang et al., 2015; 2019; Nissen et al., 2021), 2) Pre-trained Kmer embeddings (Ng, 2017; Ren et al., 2022; Han et al., 2022), and 3) Genome foundation models (Ji et al., 2021; Nguyen et al., 2023; Zhou et al., 2023). The first two methods, while straightforward, often fail to capture complex semantic relationships inherent in genomic data. Genome foundation models (Ji et al., 2021; Nguyen et al., 2023; Zhou et al., 2023), despite their successes in various genomic tasks through model fine-tuning, generally fail to develop embeddings that can discriminate certain properties like species. This is largely due to a mismatch between their pre-training objectives (Radford et al., 2019; Devlin et al., 2018) and the specific application scenarios. Our empirical analysis, as detailed in Table 1, reveals that in many scenarios, existing genome foundation models even underperform simple textual features.

In this work, we introduce DNABERT-S, a specialized genome model that harnesses the capabilities of genome foundation models to generate species-aware DNA embeddings. As depicted in Figure 1,

---

[1]Model, codes, and data will be publicly available.

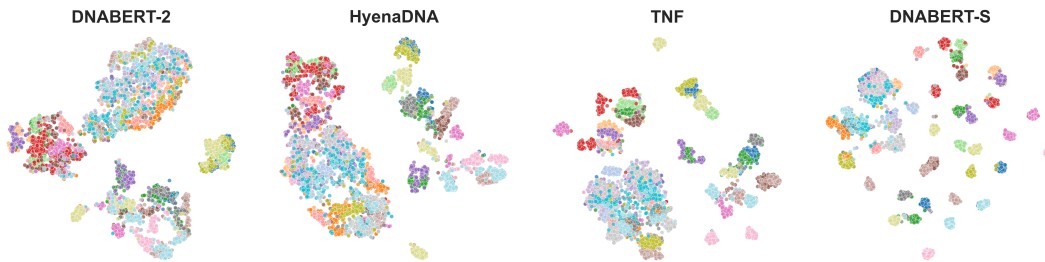

Figure 1: TSNE visualization of the DNA embeddings generated by different methods on a CAMI2 (Meyer et al., 2022) dataset with 50 different species. Each point represents an individual DNA sequence, with the color coding indicating the species affiliation. Notably, DNABERT-S demonstrates a pronounced ability to cluster and segregate different species within the embedding space.

DNABERT-S distinguishes itself from other methods by its ability to effectively cluster and separate different species within the embedding space. This enhanced performance stems from the proposed Manifold Instance Mixup (MI-Mix) loss and Curriculum Contrastive Learning ($C^2$LR) strategy. Contrastive learning enables the model to discern between similar and dissimilar DNA sequences, and curriculum learning incrementally presents more challenging training samples, fostering better learning and generalization. The training of DNABERT-S includes two phases. In the first phase, we adopt a Weighted SimCLR Chen et al. (2020); Zhou et al. (2022) training objective to encourage the model to group DNA sequences from the same species and separate DNA sequences from distinct species. In the second phase, we introduce Manifold Instance Mixup (MI-Mix), which mixes anchor instances at a randomly selected layer to create more challenging anchors for contrastive training.

To rigorously evaluate the models, we assemble a comprehensive benchmark that includes thousands of species, capturing the diversity of natural microbial communities. This benchmark incorporates complex samples from CAMI2 (Meyer et al., 2022), a leading metagenomics binning benchmark, and extensive reference genomes from Genbank (Benson et al., 2012). Thus, it includes both natural yet error-prone long-read sequences and well-curated yet potentially biased reference genomes.

We evaluate the embedding quality from entirely unsupervised problems to classification tasks with abundant labels. Experimental results indicate the effective performance of DNABERT-S. Compared to the strongest existing method, DNABERT-S doubles its performance in the clustering task and achieves better performance with only 20% of labeled data in the classification task (e.g., 2-shot v.s. 10-shot). Notably, in metagenomics binning, DNABERT-S is able to recover over 40% and 80% of species with an F1 score of over 0.5 respectively from synthetics and more realistic datasets, which is also one time more than the strongest baseline. We also show that a simple K-Nearest-Neighbors species classifier, which is trained on DNABERT-S embeddings of a small portion of each target genome, can slightly outperform a well-established traditional method MMseqs2 (Steinegger & Söding, 2017) that relies on the entire reference genomes of target species for classification.

Our contribution can be summarized as follows: 1) We demonstrate the effectiveness of genome foundation models in learning DNA embeddings, opening new avenues for various genomics research problems; 2) We introduce DNABERT-S, a model that develops distinctly better embeddings for species differentiation; 3) We propose the Curriculum Contrastive Learning ($C^2$LR) strategy with the Manifold Instance Mixup (MI-Mix) loss; 4) We publish a large-scale evaluation benchmark.

## 2 BACKGROUND

This study aims to build a species-aware DNA embedding model that maps each DNA sequence as a fixed-size numerical vector in an embedding space, where sequences from distinct species are naturally clustered and segregated. A DNA sequence is essentially a sentence composed of four unique characters: A, T, C, and G.

Existing works highly rely on descriptive textual features (Kang et al., 2015; 2019; Nissen et al., 2021) and pre-trained K-mer embeddings (Ng, 2017; Ren et al., 2022; Han et al., 2022) to compute DNA embeddings. A representative descriptive textual feature is Tetra-Nucleotide Frequency (TNF), a 256-dimensional vector where each position represents the frequency of each unique 4-mer (e.g., TTCA, AACG) in the input DNA sequence. Despite its simplicity and effectiveness, this method is limited since it singly relies on the 4-mer frequency and is not trainable to better fit downstream applications. Besides, our empirical analysis also suggests that a naive trainable model based on TNF, such as a Variational AutoEnoder (Kingma & Welling, 2013) with TNF as input, results in worse

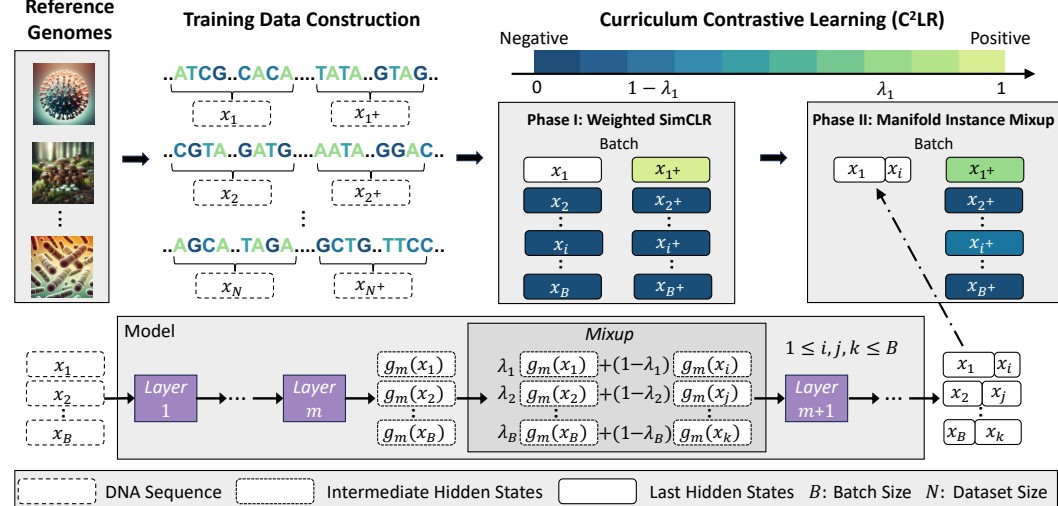

Figure 2: Overview of DNABERT-S's training process. We construct training data from massive reference genomes and train DNABERT-S with the proposed Curriculum Contrastive Learning ($C^2$LR) strategy that progressively provides more challenging contrastive anchors to the model in two different phases. We propose the Manifold Instance Mixup (MI-Mix) objective that mixes the intermediate hidden states of different inputs to construct more challenging contrastive anchor.

embeddings compared to TNF. With the success of Word2Vec (Mikolov et al., 2013), pre-trained Kmer embeddings have gained popularity in computing DNA embeddings for various applications (Ng, 2017; Ren et al., 2022; Han et al., 2022). However, the emergence of deep learning advancements such as ELMo and BERT (Peters et al., 2018; Devlin et al., 2018) highlights the limitations of static word embeddings compared to contextual embeddings produced by foundation models.

Recently, genome foundation models such as DNABERT-2 and HyenaDNA have demonstrated their effectiveness in genome analysis (Ji et al., 2021; Dalla-Torre et al., 2023; Nguyen et al., 2023; Zhou et al., 2023). However, these models do not naturally develop discriminative embeddings, largely due to the discrepancy between their language-modeling training objectives and the goal of segregating sequences in the embedding space (Li et al., 2020). To leverage the power and potential of genome foundation models, we tailor a contrastive learning method (Reimers & Gurevych, 2019; Gao et al., 2021; Chen et al., 2020; Lee et al., 2020) for DNA embedding learning by introducing the curriculum contrastive learning ($C^2$LR) strategy with the Manifold Instance Mixup (MI-Mix) training objective.

## 3 MODEL

The proposed Curriculum Contrastive Learning ($C^2$LR) splits the training process into two phases, gradually creating more challenging anchors. In phase I, we apply an effective contrastive learning method named Weighted SimCLR based on SimCLR and Hard-Negative sampling strategy (Sec. 3.1). In phase II, we propose the Manifold Instance Mixup method which creates more challenging anchors by mixing intermediate hidden states of inputs in a randomly selected hidden layer of the model (Sec. 3.2). Implementation details of DNABERT-S are presented in Sec. 3.3.

**Notation:** Let $x_i$ be an input sample. Given a batch $\{(x_i, x_{i^+})\}_{i=1}^B$, where $B$ is the batch size and $(x_i, x_{i^+})$ represents a pair of similar samples (a.k.a., positive pair). In our setting, a positive pair $(x_i, x_{i^+})$ represents two non-overlapping DNA sequences from the same genome. Let $f(\cdot)$ define the embedding model, which takes $x_i$ as input and computes fixed-size embedding $f(x_i)$.

### 3.1 WEIGHTED SIMCLR

SimCLR (Chen et al., 2020) is a simple and effective framework for contrastive learning. For an anchor $x_i$ in batch $\{(x_i, x_{i^+})\}_{i=1}^B$, SimCLR treats all the other $2B - 2$ samples in the same batch as negative samples. It encourages the model to increase the anchor's similarity with its positive sample $x_{i^+}$ and reduces its similarity with the negative samples. It treats all negative samples equally. However, recent works (Zhang et al., 2021) have suggested that hard negatives that are closer to the anchor in the representation space offer more informative learning contrasts. Therefore, Weighted SimCLR (Zhang et al., 2021) gives higher weights to negative samples that are closer to the anchor. To align with subsequent sections, we introduce the virtual labels. The label for $(x_i, x_{i^+})$ is $v_i \in \{0, 1\}^B$,

where $v_{i,i} = 1$ indicates positive samples, and $v_{i,j \neq i} = 0$ indicates negative samples. The Weighted SimCLR loss for $x_i$ is defined as:

$$\ell(f(x_i), v_i) = - \sum_{n=1}^{B} v_{i,n} \log \frac{\exp\left(\mathrm{s}\left(f(x_i), f(x_{n^+})\right)/\tau\right)}{\sum_{j \neq i} \alpha_{ij} \exp\left(\mathrm{s}\left(f(x_i), f(x_j)\right)/\tau\right)}, \tag{1}$$

where $\tau$ denotes the temperature and $\mathrm{s}(\cdot, \cdot)$ denotes the cosine similarity between two inputs. Weights $\alpha_{ij}$ denotes the relative importance of $x_j$ for optimizing the contrastive loss of anchor $x_i$ among all the $2B - 2$ negative samples. A negative sample that is closer to the anchor receives a higher weight. We set $\alpha_{ii^+} = 1$ and compute $\alpha_{ij}$ as:

$$\alpha_{ij} = \frac{\exp\left(\mathrm{s}\left(f(x_i), f(x_j)\right)/\tau\right)}{\frac{1}{2B-2} \sum_{k \neq i, i^+} \exp\left(\mathrm{s}\left(f(x_i), f(x_k)\right)/\tau\right)}.$$

For each positive pair $(x_i, x_{i^+})$, Weighted SimCLR respectively takes $x_i$ and $x_{i^+}$ as the contrastive anchors to calculate the contrastive loss. It defines the loss $\ell(f(x_{i^+}), v_i)$ for $x_{i^+}$ by exchanging the roles of instances $\{x_i\}_{i=1}^{B}$ and $\{x_{i^+}\}_{i=1}^{B}$ in Eq. (1) respectively. Therefore, the Weighted SimCLR loss on the entire batch is defined as:

$$\mathcal{L} = \frac{1}{2B} \sum_{i=1}^{B} (\ell(f(x_i), v_i) + \ell(f(x_{i^+}), v_i)). \tag{2}$$

## 3.2 Curriculum Contrastive Learning (C²LR)

In this part, we introduce our curriculum contrastive learning (C²LR) method. Curriculum learning is an effective training method that first presents easy training batches and then progresses to more challenging ones (Hacohen & Weinshall, 2019). Recent studies have successfully applied this technique to both positive pairs (Ye et al., 2021; Roy & Etemad, 2023) and negative pairs (Chu et al., 2021) in contrastive learning. We take this approach a step further by applying it to contrastive anchors, effectively using it for both types of pairs at the same time.

As shown in Figure 2, our C²LR method includes two training phases, with anchors becoming progressively more challenging. In phase I, we use the Weighted SimCLR introduced in Sec. 3.1. In phase II, we propose the Manifold Instance Mixup (MI-Mix) method to mix up anchor instances in a random hidden layer, motivated by the instance mixup (i-Mix) method (Lee et al., 2020).

The i-Mix method mixes anchors at the input layer to create more challenging positive and negative pairs. It only uses the samples from $\{x_i\}_{i=1}^{B}$ as anchors and only considers the positive and negative samples from $\{x_{i^+}\}_{i=1}^{B}$. Otherwise, it nearly doubles the memory or training time compared to the Weighted SimCLR method in Sec. 3.1 (see Appendix D for details). To perform mixup within the anchor space, i-Mix first shuffles $\{(x_i, v_i)\}_{i=1}^{B}$ to generate $\{(\hat{x}_i, \hat{v}_i)\}_{i=1}^{B}$, i.e. a random permutation of $\{(x_i, v_i)\}_{i=1}^{B}$. Then for each anchor $(x_i, v_i)$, i-Mix mixes it with $(\hat{x}_i, \hat{v}_i)$ through weighted sum. The mixing weight $\lambda_i$ is drawn from Beta$(\alpha, \alpha)$, where $\alpha$ is a hyperparameter.

Despite i-Mix's effectiveness on continuous data such as images and speeches, directly mixing DNA sequences may avoid biological plausibility. Thus, we proposed to instead mix hidden representations of DNA sequences at a deeper layer, which essentially combines more abstract, higher-level features of the sequences. We call it Manifold Instance Mixup, inspired by Verma et al. (2019). Concretely, we denote the model $f(\cdot)$ as $f(x) = f_m(g_m(x))$. Here, $g_m(\cdot)$ maps input data to the intermediate hidden states at layer $m$, and $f_m(\cdot)$ maps these intermediate hidden states to the output $f(x)$.

The Manifold Instance Mixup includes four steps. First, we uniformly select a random layer $m$ from a set of eligible layers $\mathcal{S}$ in the model, like one of the encoder layers in DNABERT-S. Second, for a batch of anchors $\{(x_i, v_i)\}_{i=1}^{B}$, we process them up to layer $m$, resulting in a batch of intermediate hidden states $\{(g_m(x_i), v_i)\}_{i=1}^{B}$. Third, we shuffle $\{(g_m(x_i), v_i)\}_{i=1}^{B}$ to get $\{(g_m(\hat{x}_i), \hat{v}_i)\}_{i=1}^{B}$ and mix them up. This produces the mixed hidden states $h_i^m$ for each $(g_m(x_i), v_i)$, where $h_i^m = \lambda_i g_m(x_i) + (1 - \lambda_i)g_m(\hat{x}_i)$, and $v_i^{mix} = \lambda_i v_i + (1 - \lambda_i)\hat{v}_i$. Fourth, we feed $\{h_i^m\}_{i=1}^{B}$ through the remaining layers to get the last hidden states $\{f_m(h_i^m)\}_{i=1}^{B}$. Loss on $i$-th anchor $(x_i, v_i)$ is defined as:

$$\hat{\ell}(f_m(h_i^m), v_i^{mix}) = - \sum_{n=1}^{B} v_{i,n}^{mix} \log \frac{\exp\left(\mathrm{s}\left(f_m(h_i^m), f(x_{n^+})\right)/\tau\right)}{\sum_{j=1}^{B} \alpha_{ij^+} \exp\left(\mathrm{s}\left(f_m(h_i^m), f(x_{j^+})\right)/\tau\right)},$$

where weights $\alpha_{ii^+} = 1$ and $\alpha_{ij^+}$ is computed as:

$$\alpha_{ij^+} = \frac{\exp\left(\mathrm{s}\left(f_m(h_i^m), f(x_{j^+})\right)/\tau\right)}{\frac{1}{B-1}\sum_{k=1, k\neq i}^{B} \exp\left(\mathrm{s}\left(f_m(h_i^m), f(x_{k^+})\right)/\tau\right)}.$$

The Manifold Instance Mixup loss is defined as follows:

$$\hat{\mathcal{L}} = \frac{1}{B}\sum_{i=1}^{B} \hat{\ell}(f_m(h_i^m), v_i^{mix}). \tag{3}$$

## 3.3 IMPLEMENTATION

In the C$^2$LR method, we set temperature $\tau$ as $0.05$ and hyperparameter $\alpha$ as $1.0$. We train the model for one epoch in phase I using loss Eq. (2) and for two epochs in phase II using loss Eq. (3). We use mean pooling of the last hidden states of all the tokens as the DNA embedding. We employ the Adam optimizer (Kingma & Ba, 2014), with a learning rate of $3e-6$ and batch size of $48$. We save the model every 10000 training steps and select the best one based on the validation loss in the validation dataset. We use the pre-trained DNABERT-2 (Zhou et al., 2023) as the starting point of contrastive training. We also conduct parallel experiments with HyenaDNA (Nguyen et al., 2023). In Sec. 5.6, we show that DNABERT-2 outperforms HyenaDNA after the same contrastive training. The training of DNABERT-S takes approximately 48 hours on 8 NVIDIA A100 80GB GPUs.

## 4 DATA

In this section, we introduce the dataset we used for DNABERT-S training and evaluation.

**Training** Each training sample of DNABERT-S is a pair of non-overlapping DNA sequences extracted from the same species. We focus on microbe species since the genetic diversity within this group provides a rich substrate for examining the nuances of species differentiation. The dataset is constructed with the reference genomes from GenBank (Benson et al., 2012). We obtained 47923 pairs from 17636 viral genomes, 1 million pairs from 5011 fungi genomes, and 1 million pairs from 6402 bacteria genomes. We randomly selected 2 million pairs from the entire 2047923 pairs of DNA sequences to construct the training data. The rest pairs are treated as validation data. All the DNA sequences are 10000 bp in length.

**Evaluation** Our evaluation spans 14 long-read datasets from the Critical Assessment of Metagenome Interpretation (CAMI) II (Meyer et al., 2022) challenge benchmark and 9 synthetic datasets from reference genomes. CAMI2 is one of the most comprehensive and rigorous benchmarks for metagenomics research. The datasets in CAMI2 are designed to mimic realistic microbiome environments and include a vast array of both new and known genomes, as well as plasmids and viruses. It aligns our study with real-world ecological and biological scenarios, providing a robust and contextually relevant evaluation for the DNA embedding models. We utilize 7 datasets of long-read contigs respectively from the `Marine` and `Plant`-associated environments, where each dataset consists of 150k-200k DNA sequences belonging to about $100-750$ different species sampled from 1680 microbial genomes and 599 circular elements. We also create 9 `Synthetic` datasets by randomly extracting DNA sequences from fungi and viral reference genomes that **do not overlap** with our training data. Table 5 in Appendix B shows the statistics of the datasets we used for evaluation.

## 5 EXPERIMENTS

We evaluate the model in a series of tasks, including: 1) metagenomics binning that identifies species from a mixture of sequences from an unknown number of species; 2) species clustering given the number of species; 3) species classification with a few labeled samples; and 4) long-read classification given reference genomes. The CAMI2 datasets are highly imbalanced, while clustering algorithms and few-shot classification are often sensitive to unbalanced data. Therefore, for the clustering and classification tasks, we filtered the datasets to eliminate species with fewer than 100 sequences and only kept 100 random sequences for each species, resulting in a set of perfectly balanced datasets. For the metagenomics binning problem, to mimic real-world scenarios, we do not balance the data.

Instead, following Kang et al. (2015), we only keep DNA sequences longer than 2500bp and filter out species with fewer than 10 sequences. Furthermore, we validate the absence of data leakage issue in our evaluation datasets. Please refer to Appendix F for details.

In this section, we present experimental design and empirical results. We introduce baselines in Sec. 5.1 and respectively present the results of clustering in Sec. 5.2, metagenomics binning in Sec. 5.3, and classification in Sec. 5.4. In Sec. 5.5, we present ablation studies on $C^2$LR and the proposed Manifold Instance Mixup training objective. Comparison with an alignment-based method for species classification is presented in Appendix C.4. We also provide empirical analysis on results with error bars (Appendix C.3), scenarios with abundant training data (Appendix C.4), results on non-microbe species (Appendix C.5), different input lengths (Appendix C.6), reduced feature dimensions (Appendix C.7), various other types of tasks (e.g., genomics function prediction) and varying backbone models (Appendix C.8). For all tasks involving randomness, we perform 5 independent runs with different random seeds for each model and report the averaged results.

## 5.1 BASELINES

We compare our model with four lines of work to examine its effectiveness. **TNF**, **TNF-K**, and **TNF-VAE** are the most widely used DNA embedding methods in metagenomics binning tools (Kang et al., 2015; 2019; Nissen et al., 2021). **TNF** represents Tetra-Nucleotide Frequency, which uses the appearance frequency of each unique 4-mer ($4^4 = 256$ in total) in a DNA sequence as its embedding. **TNF-K** (Nissen et al., 2021) reduces TNF to 103-dimension with a linear kernel, which utilizes DNA characteristics to reduce the correlations among different dimensions of the original TNF feature. **TNF-VAE** trains a Variational Autoencoder (Kingma & Welling, 2013) using TNF as input to extract features. **DNA2Vec** (Ng, 2017) learns pre-trained K-mer embedding. We set $K = 4$ to make it directly comparable with TNF and use the average of the 4-mer embeddings as the DNA embedding. **DNABERT-2** (Zhou et al., 2023), **HyenaDNA** (Nguyen et al., 2023), and **NT-v2 (Nucleotide Transformer-v2)** (Dalla-Torre et al., 2023) are representative genome foundation models. We use the average of the last hidden states as the DNA embedding. For evaluations, we utilized the respective pre-trained models from Huggingface ModelHub, specifically *zhihan1996/DNABERT-2-117M*, *LongSafari/hyenadna-medium-450k-seqlen-hf*, and *InstaDeepAI/nucleotide-transformer-v2-100m-multi-species*. **DNA-Mutate**, **DNA-Dropout**, and **DNA-Double** are variants of DNABERT-S, with the same hyperparameters but different positive pair construction strategies in contrastive training. **DNA-Mutate** views the same DNA sequence before and after random mutation (i.e., swap and delete 5% of nucleotides) as a positive pair. **DNA-Dropout** passes the same DNA sequence through the embedding model (with a dropout rate 0.1) twice and views the two distinct embeddings as a positive pair. **DNA-Double** views a DNA sequence and its reverse complementary (e.g., AATTC v.s. TTAAG) as a positive pair. **Hyena-Sim** and **DNA-Sim** are variants of HyenaDNA and DNABERT-2, fine-tuned using the Weighted SimCLR loss for 3 epochs with the same training set used for DNABERT-S. For all the models mentioned above, including DNABERT-S, we provide a detailed comparison of the number of parameters, embedding dimension, inference time, and inference memory in Appendix E. We also compare the proposed curriculum contrastive learning framework and MI-Mix training objective with well-established contrastive learning methods, including **Weight SimCLR** (Zhang et al., 2021), **i-Mix** (Lee et al., 2020) and **Supervised Contrastive Learning (SupCon)** (Khosla et al., 2021). Results on different training objectives are presented in Sec. 5.5.

## 5.2 CLUSTERING

In this task, we evaluate the embedding quality by how well a standard clustering algorithm can distinguish and cluster different species based on the embedding. To reduce the effects of other factors, we assume the number of species is known in this task. For each dataset, we compute the embedding of each DNA sequence and perform K-means clustering by setting the `num_clusters` as the number of species that exist in this dataset. We employ the Adjusted Rand Index (ARI) as the evaluation metric. ARI is a measure of the similarity between two data clusterings, adjusted for chance, providing a normalized index that ranges from $-1$ to $1$; the higher, the better.

Table 1 shows the models' performance on clustering. As shown in the table, DNABERT-S consistently achieves the best performance on all the datasets and doubles the performance of the strongest existing method on average. Among all the baselines, TNF and its variant TNF-K achieve the best performance, explaining their wide usage in metagenomics binning. Yet, TNF's performance is heavily limited since it is not learnable. TNF-VAE represents a naive algorithm that enables learning with TNF, yet it leads to big performance degradation, potentially resulting from the large gap

Table 1: Models' performance on K-Means clustering measured by Adjusted Rand Index (ARI). DNABERT-S doubles the ARI of the strongest baseline on average.

| Model | Synthetic | | Marine | | | | | Plant | | | | | Ave. |
|---|---|---|---|---|---|---|---|---|---|---|---|---|---|
| | 0 | 1 | 0 | 1 | 2 | 3 | 4 | 0 | 1 | 2 | 3 | 4 | |
| TNF | 38.75 | 37.76 | 25.65 | 25.31 | 26.05 | 20.67 | 23.47 | 25.80 | 24.23 | 24.81 | 22.72 | 22.39 | 26.47 |
| TNF-K | 36.26 | 35.66 | 25.99 | 25.00 | 26.27 | 21.15 | 23.27 | 25.60 | 25.58 | 26.45 | 22.59 | 21.76 | 26.30 |
| TNF-VAE | 25.94 | 24.60 | 16.28 | 16.52 | 16.27 | 12.92 | 15.02 | 18.40 | 16.51 | 17.53 | 14.08 | 14.38 | 17.37 |
| DNA2Vec | 24.68 | 23.34 | 16.07 | 15.99 | 16.18 | 12.62 | 14.51 | 20.13 | 19.77 | 20.25 | 17.24 | 16.37 | 18.10 |
| HyenaDNA | 20.04 | 18.99 | 16.54 | 16.64 | 16.47 | 13.35 | 14.85 | 24.06 | 25.33 | 26.18 | 21.01 | 21.16 | 19.55 |
| NT-v2 | 8.69 | 9.63 | 4.92 | 4.74 | 5.02 | 3.68 | 4.31 | 7.00 | 6.32 | 6.37 | 5.54 | 5.42 | 5.97 |
| DNABERT-2 | 15.73 | 16.74 | 13.24 | 13.53 | 12.99 | 10.41 | 11.87 | 15.70 | 16.28 | 16.32 | 13.99 | 13.66 | 14.21 |
| DNA-Dropout | 16.64 | 16.08 | 11.89 | 11.77 | 11.89 | 9.85 | 10.31 | 16.18 | 15.41 | 16.95 | 13.53 | 13.85 | 13.70 |
| DNA-Double | 35.11 | 34.14 | 27.05 | 27.23 | 26.56 | 21.47 | 24.39 | 22.35 | 21.35 | 23.03 | 19.44 | 19.06 | 25.10 |
| DNA-Mutate | 16.55 | 16.24 | 11.40 | 11.53 | 11.34 | 9.03 | 10.02 | 14.27 | 14.13 | 16.22 | 12.01 | 11.68 | 12.87 |
| Hyena-Sim | 57.58 | 55.19 | 42.92 | 42.68 | 42.24 | 36.16 | 40.09 | 41.42 | 40.36 | 40.46 | 36.87 | 38.25 | 42.85 |
| DNA-Sim | 69.33 | 68.37 | 53.18 | 51.94 | 51.91 | 46.60 | 49.69 | 49.05 | 50.33 | 49.68 | 48.57 | 49.83 | 50.08 |
| DNABERT-S | **68.21** | **66.33** | **53.98** | **52.56** | **51.99** | **46.39** | **50.49** | **51.43** | **51.56** | **51.11** | **50.44** | **51.15** | **53.80** |

between its training objective and the specific downstream application. Similarly, pre-trained Kmer embeddings from DNA2Vec also fail to effectively cluster different species.

Existing genome foundation models training with language modeling objectives, such as HyenaDNA and DNABERT-2, despite their good performance on labeled datasets, also fail to generate representative embedding without fine-tuning. The phenomenon that pre-trained foundation models underperform descriptive textual features in generating embedding for clustering and retrieval is also observed in the field of natural language processing (Reimers & Gurevych, 2019).

Furthermore, by comparing the DNA-Dropout and DNA-Mutate with DNABERT-2, we found that those popular unsupervised positive pair methods used in contrastive learning in NLP, such as sentence swap/deletion and dropout, do not benefit DNA embedding learning. The DNA-Double, which utilizes the unique double-strain characteristics of DNA sequences, empowers DNABERT-2 to achieve a similar level of performance as TNF. Comparison between DNABERT-S and these variants indicates the importance of appropriate training data construction.

## 5.3 METAGENOMICS BINNING

Metagenomics binning is a crucial process in microbial ecology, involving the categorization of DNA sequences into groups that represent individual species. State-of-the-art metagenomics binning method (Kang et al., 2015; 2019; Nissen et al., 2021) always formulate this problem as a clustering problem with an unknown number of clusters based on the feature of each DNA sequence. The DNA sequence feature is computed by combining sequence-based DNA embedding with various other features and the clustering algorithms are often complicated and strongly correlated with the features they utilize. In our evaluation, to create a fair environment for DNA embedding benchmarking, instead of relying on any existing tool, we implement the modified K-medoid clustering algorithm proposed in Kang et al. (2015) for metagenomics binning due to its simplicity and effectiveness. Algorithm 1 describes the metagenomics binning algorithm. Following Kang et al. (2015; 2019), we formulate this problem as identifying non-overlapping clusters of DNA sequences from the entire dataset, where each cluster of sequence is considered as an identified species. We iteratively identify the densest point in the embedding space and take all the sequences that are close (determined by a learned threshold) to it as the group of the sequences that belong to the same species. The embeddings of the taken sequences are removed from the embedding space. The iteration ends as there are no regions that contain enough number sequences within the threshold. This algorithm, evaluates how well the embedding method clusters and aggregates different species within the embedding space. We then compare the predicted clusters with the true labels to count the number of species that have been successfully identified. A species is considered to be successfully identified if the F1 score of this species is over $0.5$. We compare different models by the number of species they identify with different levels of F1 scores (e.g., $0.5 - 0.6$, $0.8 - 0.9$). We only use the DNA embeddings as the feature of each DNA sequence.

Figure 3 shows the models' performance on 6 metagenomics binning datasets. As shown in the figure, similar to our observation in clustering, DNABERT-S identifies twice the number of species with

an F1 score of over $0.5$ compared to the strongest baseline, showing its great capability in tackling important real-world biology challenges. Notably, DNABERT-S identifies a large number of species with an F1 score over $0.9$. indicating its capability to accurately segregate different species in the embedding space, aligning with our observation in Figure 1. In the `Synthetic` datasets, where the sequences are error-less (extracted from reference genome) and the number of sequences in each species is more balanced, DNABERT-S recovers over $80\%$ of the species with an F1 score of over $0.5$ purely based on the DNA sequences themselves. In more realistic datasets such as `Marine` and `Plant`, where noise (e.g., error from sequences) exists in DNA sequence and species size is highly imbalanced, DNABERT-S is still able to recover $40\%$ of the species with an F1 score of over $0.5$.

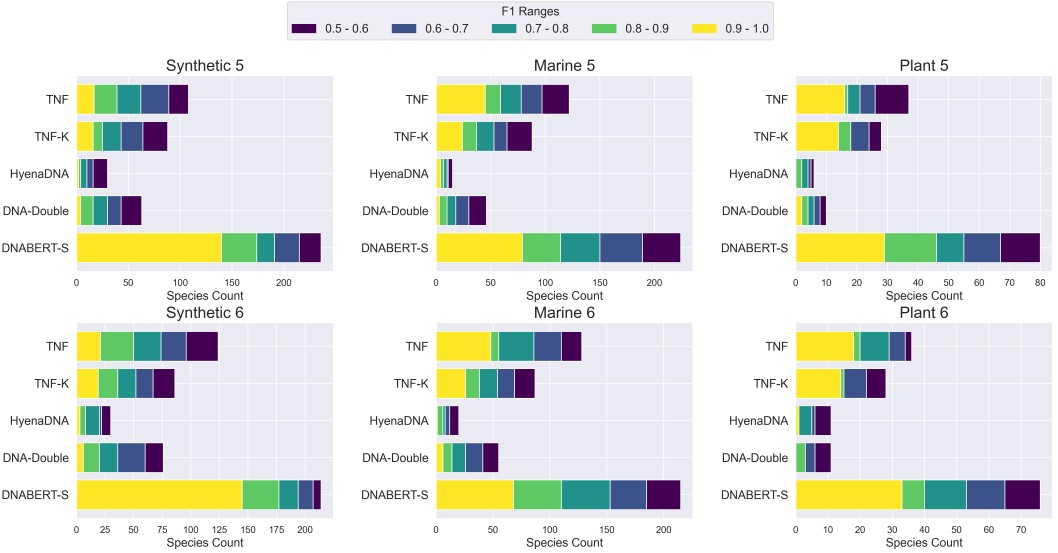

Figure 3: Metagenomics Binning Results. The bin size represents the number of unique species identified by each model and different colors represent the F1 score of the identified species. With high F1 scores, DNABERT-S identifies many more species than the baselines.

## 5.4 CLASSIFICATION

In this task, we evaluate the embedding quality by how well a simple model can classify different species based on a few labeled embeddings. We conduct experiments with a linear regression model and a non-linear multi-layer perceptron (MLP) to examine the embeddings' linear and non-linear descriptiveness. We present results on linear classification in this section. The results in non-linear settings are consistent with those in linear ones. Due to space limits, we present them in Table 6 and 7 in the Appendix C.2. As shown in Table 5, all the datasets we use for classification consist of $100$ DNA sequence for each species. We first compute the embedding of each DNA sequence with each model. In each evaluation run, we independently select $80$ embeddings from each species to form the test set. For the rest DNA sequences, we respectively sample $1, 2, 5, 10$, and $20$ embeddings from each species to form the training set. A Logistic Regression model is trained on the training set and evaluated on the test set. We use the macro F1 score as the evaluation metric.

Figure 4 shows the models' performance on 6 datasets. The results for the remaining 6 datasets are consistent and are presented in Appendix C.1. We also provide detailed results for all baselines on all 12 datasets in Table 14. As shown in the figure, DNABERT-S consistently achieves the best performance. DNABERT-S achieves better performance than the strongest baseline with only $20\%$ of training data. For example, with only 2 training samples per category, DNABERT-S achieves higher F1 scores than the strongest baseline with $10$ training samples. With the same amount of training samples, DNABERT-S outperforms the baselines by a large gap. Notably, in the Synthetic datasets, where none of the species are seen during the contrastive training, a linear model trained with DNABERT-S embeddings achieves an F1 score of over $0.8$ in 200 classes classification with only 5 labeled samples in each species, showing DNABERT-S's capability in generalizing well on unseen data. To further validate the generalizability of DNABERT-S on more distinct datasets, we compile three datasets, including genomes from invertebrate, protozoa, and mammalian species, that

are largely different from the training species (microbial genomes) of DNABERT-S. As illustrated in Sec. C.5, DNABERT-S also achieves good performance in these datasets.

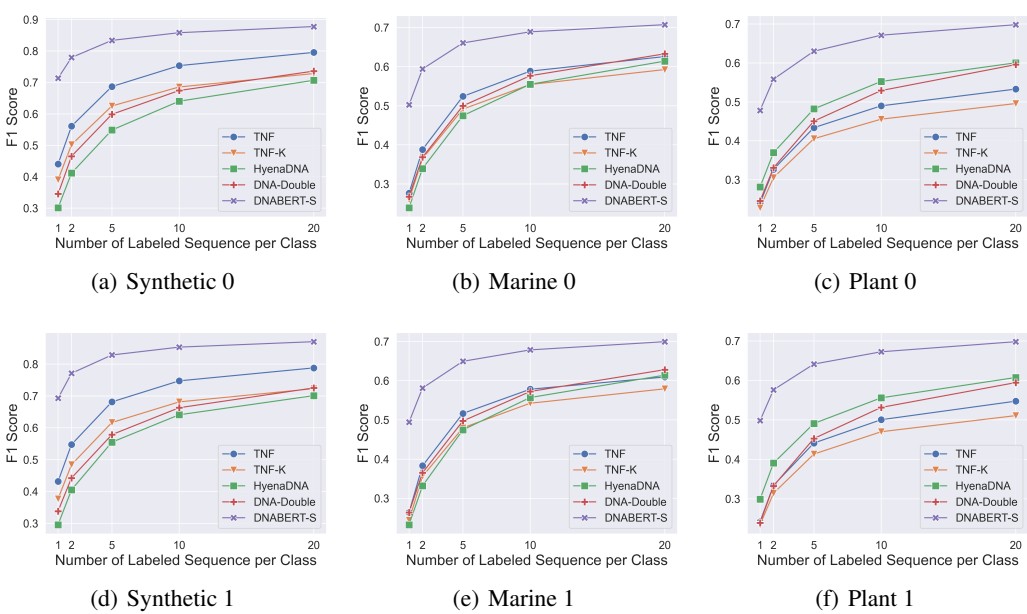

Figure 4: Model's performance of species classification with varying numbers of training samples on 6 datasets. Results on other 6 datasets are consistent and are presented in Figure 5.

## 5.5 ABLATION STUDY

In this section, we present our ablation studies on DNABERT-S. We perform the ablation study on CAMI2 datasets with both clustering and classification. To validate the effectiveness of curriculum learning, we compare DNABERT-S with three of its variants, each of which is trained purely with the Weight SimCLR (Zhang et al., 2021), i-Mix (Lee et al., 2020), SupCon (Khosla et al., 2021), and our proposed Manifold Instance Mixup (MI-Mix) loss. To examine the effectiveness of MI-Mix, we also compare it with a variant trained with the curriculum contrastive method that replaces MI-Mix with i-Mix in the second phase. All the variants are trained with the same data and hyperparameters.

As shown in Table 2, our curriculum learning strategy that combines Weighted SimCLR and MI-Mix achieves the best performance. Our method outperforms both variants that are trained purely with Weighted SimCLR and MI-Mix loss, showing the effectiveness of our proposed curriculum contrastive learning strategy. Moreover, the comparison among the four variants that are trained with a single loss function, including Weight SimCLR, i-Mix, and SupCon, indicates the effectiveness of MI-Mix in learning DNA embeddings.

Table 2: Ablation study on the Curriculum Contrastive Learning ($C^2$LR) and Manifold Instance Mixup (MI-Mix). -: Performance difference to W. SimCLR + MI-Mix.

| Training Objective | Clustering | Classification |
|---|---|---|
| W. SimCLR + MI-Mix | 51.11 | 60.88 |
| W. SimCLR + i-Mix | −3.46 | −3.56 |
| only W. SimCLR | −1.13 | −1.17 |
| only MI-Mix | −0.66 | −0.42 |
| only i-Mix | −5.25 | −4.76 |
| only SupCon | −6.75 | −3.83 |

## 5.6 SELECTION OF BACKBONE MODEL

This subsection delineates a comparative analysis of existing genome foundation models in the context of DNA embedding generation. We evaluated four renowned models: DNABERT (Ji et al., 2021), DNABERT-2 (Zhou et al., 2023), Nucleotide Transformer (Dalla-Torre et al., 2023), and HyenaDNA (Nguyen et al., 2023). Notably, DNABERT and the Nucleotide Transformer exhibit strict input sequence length limitations of 512 and 6144 (V1) or 12288 (V2) base pairs, respectively. Conversely, DNABERT-2 and HyenaDNA do not impose such constraints. Considering the potentially extensive length of genome sequences in metagenomics binning, our preliminary experiments focused solely on DNABERT-2 and HyenaDNA.

We train both models on our pre-training datasets with the same set of hyperparameters for 3 epoch. We save checkpoints periodically and select the best checkpoint based on the models' validation loss on the validation set. Since HyenaDNA, in general, requires larger learning rates than DNABERT-2, we train it with three different learning rates ($3e-4$, $3e-5$, and $3e-6$) and select the one that works best. For DNABERT-2, we only train it once with a learning rate of $3e-6$. To avoid the impact of other factors, such as the schedule of curriculum learning, we train both models with the Weighted SimCLR loss only in the entire training process. We evaluate the models before and after contrastive training on our evaluation benchmark.

Table 3 presents the performance of both pre-trained DNABERT-2 and HyenaDNA, with and without contrastive training, in K-means clustering evaluations. DNABERT-2, despite underperforming without contrastive training, demonstrated superior performance compared to HyenaDNA post-training, highlighting its efficacy in learning effective DNA embeddings. Similar trends are observed in few-shot species classification scenarios, as detailed in Table 4. While DNABERT-2 initially exhibited subpar performance, it largely improved post-contrastive training, achieving the best results. Thus, DNABERT-2 was chosen as the backbone model for DNABERT-S.

Table 3: Performance of DNABERT-2 and HyenaDNA on K-Means clustering measured by Adjusted Rand Index (ARI). $\Delta$: the model's performance improvement after contrastive training,

| | Synthetic | | Marine | | | | | Plant | | | | | Ave. |
|---|---|---|---|---|---|---|---|---|---|---|---|---|---|
| **Dataset ID** | **0** | **1** | **0** | **1** | **2** | **3** | **4** | **0** | **1** | **2** | **3** | **4** | |
| **DNABERT-2 w/o** | 15.73 | 16.74 | 13.24 | 13.53 | 12.99 | 10.41 | 11.87 | 15.70 | 16.28 | 16.32 | 13.99 | 13.66 | 14.21 |
| **DNABERT-2 w/** | **69.33** | **68.37** | **53.18** | **51.94** | **51.91** | **46.60** | **49.69** | **49.05** | **50.33** | **49.68** | **48.57** | **49.83** | **50.08** |
| $\Delta$ | 53.61 | 51.63 | 39.94 | 38.41 | 38.93 | 36.19 | 37.82 | 33.35 | 34.05 | 33.36 | 34.57 | 36.17 | 39.00 |
| **HyenaDNA w/o** | 20.04 | 18.99 | 16.54 | 16.64 | 16.47 | 13.35 | 14.85 | 24.06 | 25.33 | 26.18 | 21.01 | 21.16 | 19.55 |
| **HyenaDNA w/** | 57.58 | 55.19 | 42.92 | 42.68 | 42.24 | 36.17 | 40.10 | 41.42 | 40.36 | 40.46 | 36.87 | 38.25 | 42.85 |
| $\Delta$ | 37.54 | 36.20 | 26.38 | 26.04 | 25.77 | 22.82 | 25.24 | 17.36 | 15.03 | 14.28 | 15.86 | 17.09 | 23.30 |

Table 4: Performance of DNABERT-2 and HyenaDNA on few-shot classification measured by Macro F1 score. $\Delta$: the model's performance improvement after contrastive training,

| | Synthetic | | | | Marine | | | | Plant | | | | Ave. |
|---|---|---|---|---|---|---|---|---|---|---|---|---|---|
| **Num Shots** | **1** | **2** | **5** | **10** | **1** | **2** | **5** | **10** | **1** | **2** | **5** | **10** | |
| **DNABERT-2 w/o** | 24.43 | 34.81 | 48.93 | 58.58 | 19.50 | 28.45 | 40.64 | 48.98 | 21.04 | 28.16 | 38.50 | 45.46 | 36.46 |
| **DNABERT-2 w/** | **72.02** | **78.63** | **84.55** | **86.99** | **48.23** | **57.90** | **64.80** | **67.94** | **44.97** | **52.35** | **60.35** | **64.63** | **65.28** |
| $\Delta$ | 47.60 | 43.83 | 35.62 | 28.41 | 28.73 | 29.45 | 24.16 | 18.96 | 23.92 | 24.20 | 21.85 | 19.17 | 28.82 |
| **HyenaDNA w/o** | 30.13 | 41.18 | 54.86 | 64.03 | 23.92 | 33.94 | 47.47 | 55.50 | 28.15 | 36.97 | 48.20 | 55.24 | 43.30 |
| **HyenaDNA w/** | 59.58 | 67.79 | 74.62 | 78.53 | 43.60 | 53.70 | 62.00 | 65.55 | 43.46 | 52.12 | 59.40 | 62.80 | 60.26 |
| $\Delta$ | 29.45 | 26.62 | 19.76 | 14.50 | 19.68 | 19.76 | 14.52 | 10.05 | 15.31 | 15.15 | 11.20 | 7.55 | 16.96 |

## 6 CONCLUSION

We introduce DNABERT-S, a model tailored for species-aware DNA embeddings, which is empowered by the proposed Manifold Instance Mixup (MI-Mix) training objective and the Curriculum Contrastive Learning (C$^2$LR) strategy. We perform extensive experiments on 23 datasets across thousands of different species and a variety of challenging tasks, including species clustering, classification, and metagenomics binning, to demonstrate the DNABERT-S's capability of species differentiation. Furthermore, we conduct a series of experiments on the training objective, backbone model selection, impacts of sequence length, and feature dimension to provide empirical insights for DNA embedding learning. We also compare DNABERT-S with traditional technique like MMSeq2 in the problem of species differentiation and demonstrate DNABERT-S can achieves slightly better performance with less amount of labeled samples. We envision DNABERT-S to potentially change the way genomic problems are approached from an embedding perspective. The primary limitation of DNABERT-S lies in its high computational demands, a common trait among deep learning models, when compared to more traditional, lightweight methods like Tetranucleotide Frequencies (TNF).

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

## A  ALGORITHM FOR METAGENOMICS BINNING

Algorithm 1 describes the unsupervised clustering algorithm we used for metagenomics binning, where $\mathrm{s}(E_i, E_j)$ represents the cosine similarity of two vectors $E_i$ and $E_j$. **Selection of threshold $\gamma$.** As shown in Algorithm 1, the threshold $\gamma$ is the most important hyperparameter that greatly impacts the final binning results. A high threshold results in small and dense clusters while a low threshold results in large yet sparse clusters. Since different models generate embeddings with distinct distributions, a fixed threshold (e.g., 0.9) could be too high for one model yet too low for another one. In practice, massive hyperparameter searches are needed to determine the best threshold for each model on different datasets. Due to the large size of our experiments and the various types of models we used, an automatic way is needed to fairly choose the threshold for each model on each dataset. For each metagenomics binning dataset, we use the dataset from the same source (e.g., Marine) as it with ID 0 to compute a threshold for each model on it, Specifically, we generate embeddings for each DNA sequence in the dataset and compute the similarities between each DNA sequence and its species center (i.e., the average of all the DNA sequence belongs to this species). The 70 percentile of all the similarities is used as the threshold. **Other hyperparameters.** We set minimum bin size $m = 10$, number of steps $Z = 1000$, and number of iterations $T = 3$. We also experimented with $T = 3, 4, 5$ and $60, 70, 80, 90$ percentile of all the similarities is used as the threshold $\gamma$, and found that the results are robust to these hyperparameters.

---

**Algorithm 1** Modified $K$-Medoid Clustering

---

1: **Input:** threshold $\gamma$, minimum bin size $m$, embeddings $\boldsymbol{E} \in \mathbb{R}^{N \times d}$, number of steps $Z$, number of iterations $T$
2: **Initialize:** predictions $\boldsymbol{p} \in \mathbb{R}^N, p_i = -1$ for $i = 1, \ldots, N$, similarity matrix $\boldsymbol{S} = \boldsymbol{E}\boldsymbol{E}^\top$ with $S_{ij} = 0$ if $S_{ij} < \gamma$, density vector $\boldsymbol{d} \in \mathbb{R}^N$ with $d_i = \sum_{j=1}^N S_{ij}$
3: **for** step $z = 1$ to $Z$ **do**
4:     Select seed index $s = \arg\max_{s'} d_{s'}$ and corresponding seed $E_s$
5:     **for** iteration $t = 1$ to $T$ **do**
6:         Find neighborhood indices $\mathcal{I}$ of $E_s$ where $\mathrm{s}(E_i, E_s) > \gamma$ and $p_i = -1$ for each $i \in \mathcal{I}$
7:         Update seed: $E_s \leftarrow \frac{1}{|\mathcal{I}|} \sum_{i \in \mathcal{I}} E_i$
8:     **end for**
9:     Set $p_i \leftarrow z, d_i \leftarrow 0$ for each $i \in \mathcal{I}$
10:     Set $d_x \leftarrow d_x - \sum_{i \in \mathcal{I}} S_{xi}$ for each $x \in [1, 2, \ldots, N]$
11: **end for**
12: **for** step $z = 1$ to $Z$ **do**
13:     Find indices $\mathcal{I}$ where $p_i = z$ for each $i \in \mathcal{I}$
14:     **if** $|\mathcal{I}| < m$ **then**
15:         Set $p_i \leftarrow -1$ for each $i \in \mathcal{I}$
16:     **end if**
17: **end for**
    **Return:** predictions $\boldsymbol{p}$

---

## B  DATA STATISTICS OF EVALUATION BENCHMARK

This section details the comprehensive statistics of the 18 datasets utilized for evaluating various DNA embedding models, as summarized in Table 5. For tasks involving clustering and classification, each dataset encompasses between 93 to 499 distinct species. From each species, 100 DNA sequences are sampled. These sequences vary in length, ranging from 2,000 to 20,000 base pairs. In the case of metagenomics binning, the datasets exhibit an unbalanced distribution of sequences across different species. Specifically, the number of sequences per species varies significantly, ranging from as few as 10 to as many as 4,599.

Our benchmark contains assets from GenBank (Benson et al., 2012) (license: Creative Commons Attribution Non-Commercial License http://creativecommons.org/licenses/by-nc/2.0/uk/) and CAMI2 (Meyer et al., 2022) (license: Creative Commons Attribution 4.0 International License http://creativecommons.org/licenses/by/4.0/). It is worth noting that both GenBank and CAMI2 preprocess the RNA sequences into DNA equivalents by replacing

U with T. Thus, although we did not explicitly analyze RNA sequences, many RNA viruses are considered in both model training and evaluation.

Table 5: Data statistics of the datasets for the DNA embedding evaluation. This table presents the sampling source, ID, number of sequences, number of sequences, and the minimum / maximum / medium values of the sequence lengths and number of sequences in each species. We use the same set of balanced datasets for clustering and classification and another set of datasets for metagenomics binning.

| Tasks | Source | ID | Species | Sequences | Sequence Length | Num. Per Species |
|---|---|---|---|---|---|---|
| **Unsupervised** | Marine | 0 | 326 | 32600 | 2k / 20k / 7.6k | 100 / 100 / 100 |
| **Clustering** | Marine | 1 | 375 | 37500 | 2k / 20k / 8.2k | 100 / 100 / 100 |
| **&** | Marine | 2 | 361 | 36100 | 2k / 20k / 8.5k | 100 / 100 / 100 |
| **Few-Shot** | Marine | 3 | 499 | 49900 | 2k / 20k / 6.8k | 100 / 100 / 100 |
| **Classification** | Marine | 4 | 360 | 36000 | 2k / 20k / 7.1k | 100 / 100 / 100 |
| | Plant | 0 | 108 | 10800 | 2k / 20k / 6.6k | 100 / 100 / 100 |
| | Plant | 1 | 100 | 10000 | 2k / 20k / 6.4k | 100 / 100 / 100 |
| | Plant | 2 | 93 | 9300 | 2k / 20k / 6.2k | 100 / 100 / 100 |
| | Plant | 3 | 129 | 12900 | 2k / 20k / 5.5k | 100 / 100 / 100 |
| | Plant | 4 | 129 | 12900 | 2k / 20k / 5.7k | 100 / 100 / 100 |
| **(Microbe)** | Synthetic | 0 | 200 | 20000 | 10k / 10k / 10k | 100 / 100 / 100 |
| | Synthetic | 1 | 200 | 20000 | 10k / 10k / 10k | 100 / 100 / 100 |
| **(Mammalian,** | Synthetic | 2 | 210 | 21000 | 10k / 10k / 10k | 100 / 100 / 100 |
| **Invertebrate,** | Synthetic | 3 | 210 | 21000 | 10k / 10k / 10k | 100 / 100 / 100 |
| **Protozoa)** | Synthetic | 4 | 210 | 21000 | 10k / 10k / 10k | 100 / 100 / 100 |
| | Synthetic | 5 | 323 | 37278 | 10k / 10k / 10k | 31 / 200 / 111 |
| | Synthetic | 6 | 249 | 28206 | 10k / 10k / 10k | 30 / 199 / 114 |
| **Metagenomics** | Marine | 5 | 515 | 119465 | 2.5k / 20k / 4.3k | 10 / 841 / 201 |
| **Binning** | Marine | 6 | 527 | 125194 | 2.5k / 20k / 4.4k | 10 / 915 / 223 |
| **(Microbe)** | Plant | 5 | 181 | 71642 | 2.5k / 20k / 3.7k | 10 / 4293 / 190 |
| | Plant | 6 | 196 | 68426 | 2.5k / 20k / 3.7k | 10 / 4599 / 116 |
| **Classification** | Synthetic | 7 | 200 | 95309 | 10k / 10k / 10k | 452 / 600 / 600 |
| **(Microbe)** | Synthetic | 8 | 200 | 95475 | 10k / 10k / 10k | 442 / 600 / 600 |

## C MORE EXPERIMENTAL RESULTS

In this section, we provide additional experimental analysis. In Sec. C.1, we present the performance of the models on species classification using a linear regression model across 6 additional datasets not covered in Sec. 5.4. In Sec. C.2, we present the results of our investigation into the non-linear descriptiveness of embeddings by conducting experiments using logistic regression or a non-linear multi-layer perceptron (MLP). In Sec. C.3, we present detailed results on species clustering and few-shot classification on DNABERT-S and the most competitive baseline models. In Sec. C.4, we validate the effectiveness of DNABERT-S in situations where abundant labeled data is available by comparing it with MMseqs2 (Steinegger & Söding, 2017). In Sec. C.5, we show the performance of DNABERT-S to distinguish genomics sequences from species that are largely different from the ones in the training set. In Sec. C.6, we delve into the influence of DNA sequence length on the performance of DNABERT-S. In Sec. C.7, we investigate how changes in embedding dimensions affect the performance of DNABERT-S. In Sec. C.8, we evaluate the impact of species-aware embedding on other types of genomics analysis tasks, like genomics function prediction tasks.

### C.1 REMAINING RESULTS ON SPECIES CLASSIFICATION

In this section, we present the performance of the models on species classification using a linear regression model across 6 additional datasets not covered in Sec. 5.4. As shown in Figure 5, the results are consistent with those shown in Figure 4. We also provide detailed results for all baselines on all 12 datasets for completeness in Table 14.

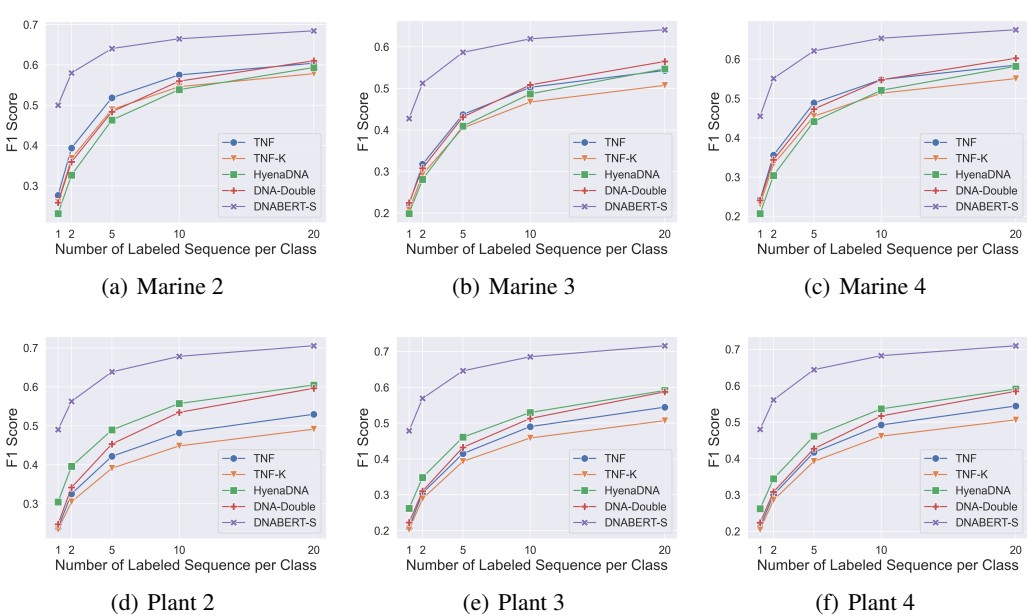

Figure 5: Results of species classification using linear regression on other 6 datasets.

## C.2 RESULTS COMPARISON WITH LINEAR AND NON-LINEAR CLASSIFIERS

In this section, we present the results of our investigation into the non-linear descriptiveness of embeddings by conducting experiments using logistic regression or a non-linear multi-layer perceptron (MLP). Table 6 and 7 show the results for three datasets: "Marine 0", "Plant 0", and "Synthetic 0". The results demonstrate that DNABERT-S consistently achieves the best performance.

Table 6: DNABERT-S's performance of **species classification** with varying numbers of training samples on datasets "Marine 0" and "Plant 0": Beyond using **logistic regression (LR)**, we also train a **multi-layer perceptron (MLP)** with non-linear activation function (ReLU). The term "**Difference**" denotes the performance gap between "DNABERT-S" and the "best-baseline". The results show that DNABERT-S embedding consistently outperforms the best-existing baseline in both linear and non-linear discriminativity.

| | Marine 0 | | | | | Plant 0 | | | | |
|---|---|---|---|---|---|---|---|---|---|---|
| **Dataset ID** | **1** | **2** | **5** | **10** | **20** | **1** | **2** | **5** | **10** | **20** |
| **LR: best-baseline** | 27.65 | 38.81 | 52.4 | 58.86 | 63.29 | 28.15 | 36.97 | 48.2 | 55.24 | 60.04 |
| **LR: DNABERT-S** | 50.25 | 59.41 | 66.07 | 68.92 | 70.75 | 47.83 | 55.83 | 63.01 | 67.12 | 69.82 |
| **LR: Difference** | **22.60** | **20.60** | **13.67** | **10.06** | **7.46** | **19.68** | **18.86** | **14.81** | **11.88** | **9.78** |
| **MLP: best-baseline** | 26.07 | 38.59 | 53.86 | 60.27 | 63.07 | 27.39 | 36.19 | 47.5 | 54.03 | 59.13 |
| **MLP: DNABERT-S** | 48.55 | 59.25 | 66.09 | 68.95 | 69.99 | 45.31 | 55.34 | 63.25 | 67.25 | 70.00 |
| **MLP: Difference** | **22.48** | **20.66** | **12.23** | **8.68** | **6.92** | **17.92** | **19.15** | **15.75** | **13.22** | **10.87** |

## C.3 RESULTS WITH ERROR BARS

In this section, we present detailed results on species clustering and few-shot classification on DNABERT-S and the most competitive baseline models. Table 8, 9, and 10 respectively show the models' mean and std on each setting across 5 random seeds. As shown in the tables, DNABERT-S consistently outperforms the baselines with small variances.

## C.4 COMPARISON WITH ALIGNMENT-BASED METHOD

While previous experiments have highlighted DNABERT-S's exceptional performance in scenarios with limited or no labeled data, this section focuses on its effectiveness in situations where abundant labeled data is available. Specifically, we aim to understand the embedding-based species

Table 7: DNABERT-S's performance of **species classification** with varying numbers of training samples on dataset "Synthetic 0": Beyond using **logistic regression (LR)**, we also train a **multi-layer perceptron (MLP)** with non-linear activation function (ReLU). The term "**Difference**" denotes the performance gap between "DNABERT-S" and the "best-baseline". The results show that DNABERT-S embedding consistently outperforms the best-existing baseline in both linear and non-linear discriminativity.

| | Synthetic 0 | | | | |
|---|---|---|---|---|---|
| Dataset ID | 1 | 2 | 5 | 10 | 20 |
| LR: best-baseline | 44.07 | 56.11 | 68.69 | 75.34 | 79.54 |
| LR: DNABERT-S | 71.36 | 77.93 | 83.37 | 85.81 | 87.77 |
| LR: Difference | **27.29** | **21.82** | **14.68** | **10.47** | **8.23** |
| MLP: best-baseline | 40.28 | 54.95 | 69.59 | 76.54 | 81.33 |
| MLP: DNABERT-S | 68.96 | 77.47 | 83.44 | 85.86 | 87.77 |
| MLP: Difference | **28.68** | **22.52** | **13.85** | **9.32** | **6.44** |

Table 8: Performance of models with error bars on "Synthetic 0" dataset. We evaluate models using K-Means clustering (Sec. 5.2) and 1/2/5/10/20-shot classification (Sec. 5.4).

| | Synthetic 0 | | | | | |
|---|---|---|---|---|---|---|
| | ARI | 1 | 2 | 5 | 10 | 20 |
| TNF | 38.18 ± 1.27 | 44.30 ± 0.97 | 56.13 ± 1.10 | 68.68 ± 0.73 | 75.24 ± 0.37 | 79.48 ± 0.07 |
| TNF-K | 36.11 ± 0.09 | 39.51 ± 0.35 | 50.26 ± 0.89 | 62.43 ± 0.41 | 68.53 ± 0.50 | 72.95 ± 0.36 |
| HyenaDNA | 20.10 ± 0.50 | 30.03 ± 0.49 | 41.21 ± 0.92 | 54.42 ± 0.73 | 63.79 ± 0.59 | 70.53 ± 0.25 |
| DNA-Double | 34.91 ± 0.90 | 34.61 ± 0.91 | 46.79 ± 0.39 | 59.92 ± 0.53 | 67.45 ± 0.20 | 73.64 ± 0.25 |
| DNABERT-S | 66.94 ± 2.07 | 71.54 ± 0.51 | 77.77 ± 0.68 | 83.12 ± 0.18 | 85.63 ± 0.18 | 87.68 ± 0.15 |

Table 9: Performance of models with error bars on "Marine 0" dataset. We evaluate models using K-Means clustering (Sec. 5.2) and 1/2/5/10/20-shot classification (Sec. 5.4).

| | Marine 0 | | | | | |
|---|---|---|---|---|---|---|
| | ARI | 1 | 2 | 5 | 10 | 20 |
| TNF | 24.78 ± 0.23 | 27.89 ± 0.95 | 38.69 ± 0.04 | 52.36 ± 0.26 | 58.95 ± 0.05 | 62.69 ± 0.10 |
| TNF-K | 25.44 ± 0.50 | 22.82 ± 0.35 | 30.06 ± 0.61 | 40.50 ± 0.14 | 45.20 ± 0.25 | 49.35 ± 0.15 |
| HyenaDNA | 16.31 ± 0.07 | 23.89 ± 0.83 | 33.59 ± 0.01 | 47.50 ± 0.39 | 55.61 ± 0.18 | 61.75 ± 0.13 |
| DNA-Double | 26.82 ± 0.45 | 26.87 ± 1.16 | 36.98 ± 0.21 | 49.86 ± 0.12 | 57.93 ± 0.26 | 63.33 ± 0.06 |
| DNABERT-S | 53.91 ± 0.22 | 50.37 ± 0.74 | 59.71 ± 0.11 | 66.03 ± 0.11 | 69.00 ± 0.19 | 70.75 ± 0.13 |

Table 10: Performance of models with error bars on "Plant 0" dataset. We evaluate models using K-Means clustering (Sec. 5.2) and 1/2/5/10/20-shot classification (Sec. 5.4).

| | Plant 0 | | | | | |
|---|---|---|---|---|---|---|
| | ARI | 1 | 2 | 5 | 10 | 20 |
| TNF | 26.10 ± 0.70 | 24.07 ± 0.40 | 32.04 ± 1.00 | 43.35 ± 0.24 | 48.92 ± 0.27 | 53.29 ± 0.39 |
| TNF-K | 25.83 ± 0.67 | 22.82 ± 0.35 | 30.06 ± 0.61 | 40.50 ± 0.14 | 45.20 ± 0.25 | 49.35 ± 0.15 |
| HyenaDNA | 24.61 ± 0.77 | 28.45 ± 0.30 | 36.46 ± 0.61 | 48.56 ± 0.30 | 55.13 ± 0.53 | 59.80 ± 0.27 |
| DNA-Double | 22.10 ± 0.46 | 24.80 ± 0.56 | 32.91 ± 0.52 | 44.82 ± 0.18 | 52.77 ± 0.48 | 59.35 ± 0.37 |
| DNABERT-S | 51.15 ± 1.13 | 48.39 ± 1.33 | 55.92 ± 0.83 | 62.97 ± 0.36 | 67.14 ± 0.41 | 69.64 ± 0.15 |

differentiation method in scenarios where reference genomes of the species to classify are available. We compare embedding-based methods with MMseqs2 (Steinegger & Söding, 2017), a leading alignment-based species classification tool.

For a fair comparison with MMseqs2, which relies on the reference genomes of each species when performing classification, we construct two datasets, each consisting of 200 distinct species. To mimic real-world setups, instead of classifying segment of reference genomes, we simulate 600 long-reads with PBSIM2 (Ono et al., 2021) from each selected species, 100 as the test set and 500 as the training set. For the embedding-based methods, such as DNABERT-S and TNF, we generate embedding for all these sequences and use K-Neareast-Neighbor (KNN) classifier with n equals to 5 for species classification. We respectively use 100, 200, 300, 400, and 500 sequences from each species to construct the training set. For the MMseqs2, we respectively use the reference genome and the simulated long-read sequences as the reference for alignment-based classification.

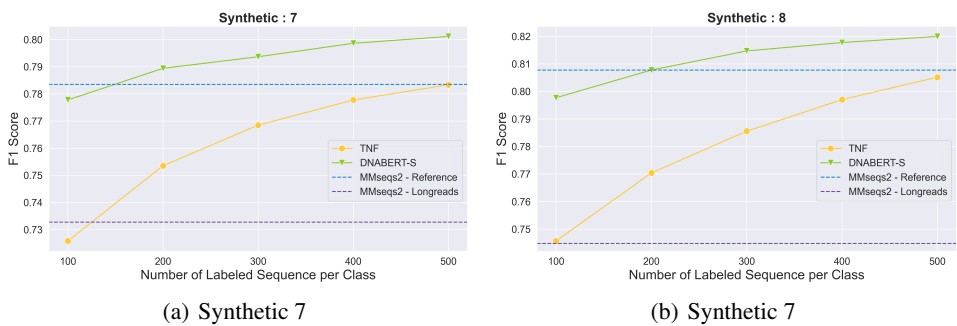

(a) Synthetic 7                        (b) Synthetic 7

Figure 6: Results on species classification when reference genomes are available.

Figure 6 the results of the models on the datasets. As shown in the figure, DNABERT-S starts to outperforms MMseqs2 with 200-300 labeled sequence from each species with a simple KNN classier, while TNF achieves comparable performance as MMseqs2 with about 500 labels sequence per species. These results indicate the potential of embedding-based methods to replace traditional alignment-based method in species classification in data abundant scenarios. Yet a fact that cannot be ignored is that DNABERT-S is much more computational cost than MMseqs2. As a comparison, MMseqs2 requires about 30 seconds on a single CPU to make predictions while DNABERT-S requires about 1 hour on 2 NVIDIA A100 GPUs to do the same thing. Therefore, there is a long way to go to fully replace alignment-based methods in high-throughput genomics analysis.

## C.5 RESULTS ON NON-MICROBE SPECIES

Since DNABERT-S is trained on microbe species (e.g., viruses, fungi, and bacteria), a natural question is whether it can distinguish genomics sequences from species that are largely different from the ones in the training set. To answer this question, we construct 3 synthetic datasets (ID: 2, 3, and 4) that include genomes from invertebrate, protozoa, and mammalian species. We randomly select 70 species from each category to achieve 210 species in total. We perform the same clustering and few-shot classification as presented in Table 1 and Figure 4.

As shown in Table 11, DNABERT-S consistently outperforms baselines across all the datasets and evaluation scenarios, indicating DNABERT-S's transferability and robustness on species that are significantly different from its training set. However, the improvements over the baselines are less significant, and the absolute scores, such as ARI in clustering and F1 in classification, are also lower than those in microbe datasets. On the one hand, there are higher genetic similarities among mammals compared to the often more significant genetic diversity found in microbes, making it more challenging to distinguish different mammalian species. On the other hand, due to the significant distinction between microbe and mammalian/protozoa species, some of the differentiation rules and markers learned from microbe genomes may not be applicable to genomes of species in other categories, which also suggests the needs of in-domain species-aware training.

## C.6 IMPACT OF SEQUENCE LENGTH

This section delves into the influence of DNA sequence length on final model performance, examined from both training and evaluation standpoints.

Table 11: Performance of models on non-microbe species. We evaluate models on each dataset using K-Means clustering (Sec. 5.2) and 1/5/20-shot classification (Sec. 5.4).

| Dataset ID | Synthetic:2 | | | | Synthetic:3 | | | | Synthetic:4 | | | |
|---|---|---|---|---|---|---|---|---|---|---|---|---|
| | ARI | 1 | 5 | 20 | ARI | 1 | 5 | 20 | ARI | 1 | 5 | 20 |
| TNF | 20.70 | 26.28 | 45.71 | 56.24 | 19.11 | 25.05 | 42.69 | 52.27 | 21.49 | 25.94 | 44.96 | 54.60 |
| TNF-K | 18.63 | 22.42 | 40.07 | 50.75 | 16.91 | 20.38 | 36.48 | 46.74 | 19.30 | 21.38 | 39.23 | 48.94 |
| HyenaDNA | 11.20 | 17.29 | 34.77 | 48.76 | 10.86 | 16.50 | 32.69 | 45.26 | 11.58 | 17.30 | 35.38 | 47.86 |
| DNABERT-2 | 9.18 | 15.91 | 33.75 | 49.00 | 8.99 | 15.28 | 31.03 | 45.17 | 9.79 | 15.38 | 33.75 | 47.78 |
| DNA-Double | 13.01 | 16.04 | 30.56 | 42.11 | 12.80 | 15.91 | 28.82 | 39.28 | 14.15 | 16.20 | 31.22 | 42.25 |
| DNABERT-S | 32.70 | 33.21 | 49.78 | 59.01 | 29.44 | 29.78 | 45.65 | 54.63 | 33.60 | 32.58 | 49.67 | 57.97 |

### C.6.1 VARYING SEQUENCE LENGTH IN TRAINING

Training with longer DNA sequences increases the need for more memory and computing power. It also means we can only use smaller batches of data at a time. Therefore, the length of the sequences is an important factor in contrastive training as it affects how much it costs to train the model. To see how different sequence lengths affect training, we did three experiments using the same data. Our training data has sequences that are 10000bp long. For experiments with shorter sequences $S$, we only used the first $S$ nucleotides of each DNA sequence. We tested sequence lengths of 500bp, 2000bp, and 10000bp, training only with Weighted SimCLR loss and starting from the pre-trained DNABERT-2 model.

Figure 7 shows the results for the three models, along with the pre-trained DNABERT-2 without contrastive training and the strongest baseline, TNF. The findings reveal that sequence length significantly influences the model's performance. Training even on short sequences, such as 500bp, leads to substantial improvements. The model trained with 500bp sequences performs nearly as well as TNF. When we increase the input sequence length from 500bp to 2000bp, there's a marked improvement in performance. A similar trend is observed when increasing the sequence length from 2000bp to 10000bp. These results highlight the importance of sequence length in training an effective model. Therefore, we decided to train our model with 10000bp sequences, despite the higher computational requirements.

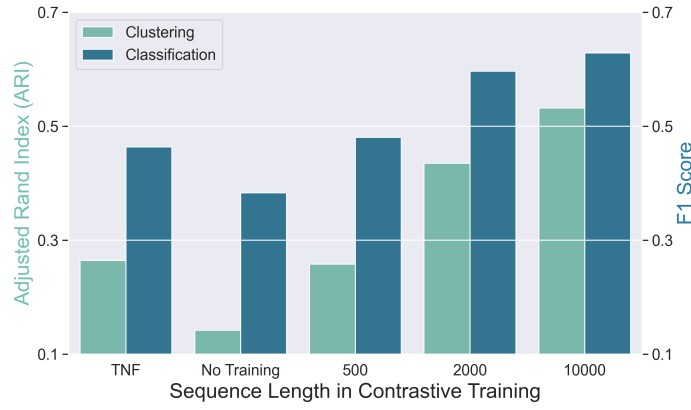

Figure 7: Performance of DNABERT-S in clustering and classification with different sequence lengths used in contrastive training.

### C.6.2 VARYING SEQUENCE LENGTH IN EVALUATION

In this part, we assess how the length of DNA sequences in evaluation impacts performance. We use two synthetic datasets for clustering and classification tasks. Each sequence in these datasets is deliberately constructed to be 10000bp long. This allows us to create a test set where all sequences have the same length. We test sequence lengths ranging from 32bp ($2^5$) to 8192bp ($2^{13}$). For each

test with different sequence lengths, we keep everything else the same, like how we split the data into training and testing sets and the settings for logistic regression.

Figure 8 presents the performance of TNF and DNABERT-S with various sequence lengths. The results show that both models significantly benefit from longer sequences. When the sequence length is less than 256bp ($2^8$), both models perform poorly in clustering and classifying samples. However, as the sequence length increases, starting from 512bp ($2^9$), DNABERT-S begins to outperform TNF. The performance gap between the two models gets bigger as the sequence length increases. These findings highlight the crucial role of sequence length in effectively differentiating between species.

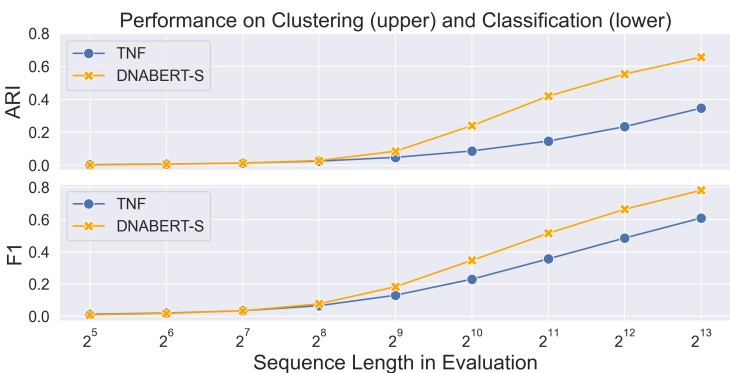

Figure 8: Performance of DNABERT-S and TNF on clustering (upper) and classification (lower) with different input sequence lengths during evaluation.

### C.7 IMPACT OF EMBEDDING DIMENSION REDUCTION

This section investigates how changes in embedding dimensions affect the performance of DNABERT-S, a key aspect influencing the scalability of DNA embeddings generated by the model. Initially, DNA embeddings for all clustering and classification datasets are computed using the pre-trained DNABERT-S. To reduce embedding dimensions, we use an average pooling layer with a consistent kernel size and stride $S$. This process effectively averages $S$ consecutive dimensions into one new dimension. We test with $S$ values of 96, 48, 24, 12, 6, 3, and 2, corresponding to reduced embedding dimensions of 8, 16, 32, 64, 128, 256, and 384, respectively.

Figure 9 illustrates DNABERT-S's performance with these varying feature dimensions, in comparison to TNF. The results demonstrate that DNABERT-S's embedding is quite resilient to dimension compression. It maintains nearly the same performance level even when reduced to 256 dimensions and only experiences a notable drop in performance when compressed to 32 dimensions. Notably, DNABERT-S still surpasses the 256-dimensional TNF feature even when its own dimensionality is reduced to just 16. This robustness to dimension reduction enhances its practical applicability in various genomic contexts.

### C.8 SPECIES-AWARE EMBEDDINGS ON GENOMICS FUNCTION PREDICTIONS

This section evaluates the impact of species-aware embedding on other types of genomics analysis tasks. Specifically, we aim to understand how the species-aware embeddings perform compared to the embedding generated by the genome foundation model without contrastive training on various types of genomics function prediction tasks. We utilize the GUE benchmark (Zhou et al., 2023), which comprises a comprehensive collection of 28 datasets covering 7 diverse tasks, such as epigenetic marks prediction, promoter prediction, and transcription factor binding site prediction. Following our established methodology, each model is respectively used to generate embeddings for each DNA sequence, and a logistic regression model is trained for classification. The Matthews Correlation Coefficient (MCC) serves as the evaluation metric. We perform experiments on both DNABERT-2 and HyenaDNA.

The results, as detailed in Table 12, show that after species-aware conservative training, DNABERT-S underperforms DNABERT-2 on 19 of the 28 datasets, with an average loss of 2.12 in the MCC. Similarly, on HyenaDNA, the one went through species-aware training underperforms the original one on 18 out of 28 datasets. As a model honed for species-aware tasks, DNABERT-S may exhibit reduced

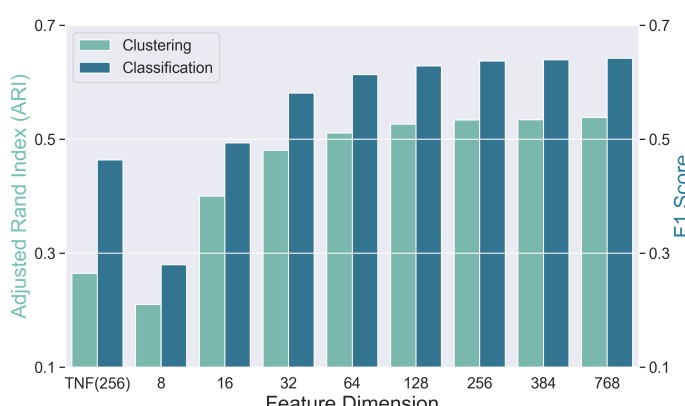

Figure 9: DNABERT-S's performance with varying embedding dimensions reduced by average pooling. DNABERT-S is robust to feature dimension reduction, and it even outperforms TNF with 16-dimensional embedding.

generalizability compared to broader genome foundation models (e.g., DNABERT-2). However, this specialized focus should not be viewed as a disadvantage. The intrinsic design of DNABERT-S—to prioritize species-specific features—may naturally limit its applicability to a broader range of tasks, a trade-off inherent to its specialized nature. For instance, when considering sequences from different species with varying functions, DNABERT-S's training objective emphasizes species-specific similarities over functional commonalities, a choice that is deliberate for its targeted application.

## D   I-MIX FOR ORIGINAL CONTRASTIVE LEARNING

The Weighted SimCLR method treat each instance $\{x_i\}_{i=1}^B$ and $\{x_{i^+}\}_{i=1}^B$ as anchors, with each anchor associated with $2B - 2$ negative samples. This section explains why integrating the i-Mix method into Weighted SimCLR (Sec. 3.1) doubles memory or training time.

For clarity, we define $x_{B+i} = x_{i^+}$. We also expand the virtual labels $v_i$, which are $2B$-dimensional, to identify the positive sample for each anchor. Here $v_{i,\tilde{i}} = 1$ and $v_{i,j\neq\tilde{i}} = 0$, $\tilde{i} = (B + i)$ mod $2B$.

When the i-Mix method treats every sample from $\{x_i\}_{i=1}^{2B}$ as anchor, it begins by shuffling $\{(x_i, v_i)\}_{i=1}^{2B}$ to generate $\{(\hat{x}_i, \hat{v}_i)\}_{i=1}^{2B}$. For each anchor $(x_i, v_i)$, it uses a simple mixup method to mix it up with $(\hat{x}_i, \hat{v}_i)$ before encoder layers of the model, using mixup coefficient $\lambda_i \sim \text{Beta}(\alpha, \alpha)$. This mixing results in:

$$(h_i^0, v_i^{mix}) = (\lambda_i x_i + (1 - \lambda_i)\hat{x}_i, \lambda_i v_i + (1 - \lambda_i)\hat{v}_i).$$

Moreover, for each anchor $x_i$, if i-Mix considers all samples from $\{x_j\}_{j=1}^{2B}\setminus\{x_i\}$ as either positive or negative samples, the model $f(\cdot)$ must process both the initial data instances $\{x_i\}_{i=1}^{2B}$ and mixed data instances $\{h_i^0\}_{i=1}^{2B}$ to generate their embeddings. Therefore, the i-Mix requires nearly twice more memory or training time compared to the method in Sec. 3.1 if using the same batch size.

Table 12: Performance of DNABERT-2 and HyenaDNA before and after species-aware contrastive training on the GUE benchmark.

| | Epigenetic Marks Prediction | | | | | |
|---|---|---|---|---|---|---|
| | **H3** | **H3K14ac** | **H3K36me3** | **H3K4me1** | **H3K4me2** | **H3K4me3** |
| **DNABERT-2** | 66.87 | **38.92** | **43.39** | 31.79 | **30.16** | **23.68** |
| **DNABERT-S** | **69.02** | 37.45 | 41.91 | **32.76** | 27.86 | 22.16 |
| **HyenaDNA w/o** | 69.54 | 30.62 | **38.95** | **33.29** | **30.90** | 21.24 |
| **HyenaDNA w/** | **69.67** | **32.06** | 38.61 | 32.29 | 30.32 | **23.21** |

| | Epigenetic Marks Prediction | | | | Promoter Detection | | |
|---|---|---|---|---|---|---|---|
| | **H3K79me3** | **H3K9ac** | **H4** | **H4ac** | **all** | **notata** | **tata** |
| **DNABERT-2** | 57.01 | **47.52** | 73.75 | 35.54 | 78.31 | 40.25 | **88.85** |
| **DNABERT-S** | **58.41** | 44.70 | **75.96** | **35.64** | **78.93** | **40.59** | 88.62 |
| **HyenaDNA w/o** | 52.33 | 44.27 | **72.93** | 29.37 | **77.06** | **53.73** | **86.25** |
| **HyenaDNA w/** | **54.99** | **44.45** | 72.61 | **29.85** | 76.56 | 45.17 | 85.14 |

| | Transcription Factor Prediction (Human) | | | | | Core Promoter Detection | | |
|---|---|---|---|---|---|---|---|---|
| | **0** | **1** | **2** | **3** | **4** | **all** | **notata** | **tata** |
| **DNABERT-2** | **62.67** | **68.58** | **54.44** | 35.67 | **62.89** | **57.33** | **46.18** | 61.48 |
| **DNABERT-S** | 60.34 | 65.28 | 47.75 | 30.54 | 59.45 | 56.52 | 39.05 | **61.87** |
| **HyenaDNA w/o** | **61.37** | **65.96** | **45.97** | 35.76 | **58.32** | **56.14** | 40.94 | **60.31** |
| **HyenaDNA w/** | 59.10 | 65.02 | 45.47 | **36.68** | 53.20 | 55.52 | **42.29** | 60.30 |

| | Transcription Factor Prediction (Mouse) | | | | | Virus | Splice | |
|---|---|---|---|---|---|---|---|---|
| | **0** | **1** | **2** | **3** | **4** | **Covid** | **Reconstruct** | **Ave.** |
| **DNABERT-2** | **37.08** | 69.56 | **67.71** | **42.26** | **34.80** | **54.50** | **24.85** | **51.28** |
| **DNABERT-S** | 31.38 | **71.13** | 56.71 | 39.85 | 28.94 | 49.84 | 23.87 | 49.16 |
| **HyenaDNA w/o** | **23.48** | **58.28** | 54.88 | 21.34 | **22.15** | 30.50 | **28.71** | **46.59** |
| **HyenaDNA w/** | 21.24 | 53.88 | **55.54** | **23.86** | 18.66 | **31.08** | 22.11 | 45.67 |

# E COMPARISON OF PARAMETERS, EMBEDDING DIMENSIONS, INFERENCE TIME, AND MEMORY

In this section, we compare the number of parameters (million), embedding dimensions, inference time (seconds), and inference memory (MB) for all models listed in Table 1. We show the results in Table 13.

Table 13: Comparison of the number of parameters (million), embedding dimensions (million), inference time (seconds), and memory (MB) for all models listed in Table 1. The symbol "-" denotes that the inference time or memory is negligible.

| Model | Num. Params (M) | Emb. Dim. | Inf. Time (Sec.) | Inf. Mem. (MB) |
|---|---|---|---|---|
| **TNF** | 0 | 256 | - | - |
| **TNF-K** | 0.026 | 768 | - | - |
| **TNF-VAE** | 3 | 103 | - | - |
| **DNA2Vec** | 0.026 | 100 | - | - |
| **HyenaDNA** | 28.2 | 256 | 11.16 | 995 |
| **Hyena-Sim** | 28.2 | 256 | 11.16 | 995 |
| **NT-v2** | 97.9 | 512 | 19.16 | 1273 |
| **DNABERT-2** | 117 | 768 | 14.27 | 3991 |
| **DNA-Dropout** | 117 | 768 | 14.27 | 3991 |
| **DNA-Double** | 117 | 768 | 14.27 | 3991 |
| **DNA-Mutate** | 117 | 768 | 14.27 | 3991 |
| **DNA-Sim** | 117 | 768 | 14.27 | 3991 |
| **DNABERT-S** | 117 | 768 | 14.27 | 3991 |

# F  Absence of Data Leakage

In this section, we consider data leakage to occur when the same species are present in both training and evaluation datasets. We validate the absence of data leakage issues in our evaluation datasets.

Our experiments utilize two categories of data: synthetic and CAMI2 datasets. For synthetic data, the data construction method ensures the absence of data leakage. We design the synthetic datasets to exclude any species present in the training data, thereby preventing potential data leakage. For the CAMI2 datasets, due to discrepancies in species annotations between CAMI2 and GenBank, direct validation was challenging. Therefore, we perform an alignment-based estimation using minimap2 (Li, 2018). We align each evaluation dataset to the training data and considered sequences with over $90\%$ alignment to the training sequences as present in the training data.

We compute the presence rate as number of presented sequences/total number of sequences. It's important to note that different species can share common or highly similar genome sequences, so a non-zero presence rate is expected in real-world scenarios. As a reference, the two synthetic datasets with non-overlapping species have presence rates of $6.88\%$ and $8.92\%$. For the CAMI2 datasets, the plant-associated ones have presence rates between $3.51\%$ and $4.98\%$, which are even lower than the synthetic datasets. The marine datasets have presence rates between $7.99\%$ and $9.45\%$, comparable to the synthetic ones. Based on these statistics, there is negligible species leakage between our training and evaluation data.

Table 14: Model's performance on species classification using linear regression with varying numbers of training samples on all the 12 datasets.

| Model | Synthetic 0 | | | | | Marine 0 | | | | | Plant 0 | | | | |
|---|---|---|---|---|---|---|---|---|---|---|---|---|---|---|---|
| | 1 | 2 | 5 | 10 | 20 | 1 | 2 | 5 | 10 | 20 | 1 | 2 | 5 | 10 | 20 |
| TNF | 44.07 | 56.11 | 68.69 | 75.34 | 79.54 | 27.65 | 38.81 | 52.4 | 58.86 | 62.59 | 24.01 | 32.69 | 43.39 | 48.99 | 53.29 |
| TNF-K | 39.06 | 50.22 | 62.52 | 68.55 | 72.82 | 25.97 | 36.47 | 49.15 | 55.44 | 59.26 | 22.83 | 30.58 | 40.55 | 45.57 | 49.58 |
| TNF-VAE | 34.23 | 47.06 | 61.44 | 69.31 | 75.02 | 23.72 | 34.02 | 47.00 | 53.88 | 58.59 | 20.63 | 28.80 | 39.38 | 45.96 | 51.10 |
| DNA2Vec | 35.85 | 46.98 | 61.54 | 69.64 | 75.26 | 24.56 | 34.04 | 47.79 | 55.36 | 60.11 | 23.98 | 31.35 | 41.46 | 47.41 | 51.96 |
| HyenaDNA | 30.13 | 41.18 | 54.86 | 64.03 | 70.69 | 23.92 | 33.94 | 47.47 | 55.50 | 61.42 | 28.15 | 36.97 | 48.20 | 55.24 | 60.04 |
| DNABERT-2 | 24.43 | 34.81 | 48.93 | 58.58 | 65.98 | 19.50 | 28.45 | 40.64 | 48.98 | 55.67 | 21.04 | 28.16 | 38.50 | 45.46 | 51.99 |
| DNA-Dropout | 21.09 | 29.39 | 40.80 | 48.38 | 54.44 | 15.42 | 21.47 | 30.99 | 38.05 | 44.06 | 19.05 | 24.78 | 33.12 | 38.99 | 44.30 |
| DNA-Double | 34.54 | 46.54 | 59.86 | 67.44 | 73.60 | 26.76 | 36.84 | 49.98 | 57.68 | 63.29 | 24.56 | 33.09 | 45.06 | 52.91 | 59.57 |
| DNA-Mutate | 21.27 | 29.92 | 41.78 | 50.31 | 57.20 | 15.70 | 21.74 | 31.86 | 39.32 | 45.95 | 18.16 | 24.40 | 33.58 | 40.23 | 46.09 |
| Hyena-Sim | 59.58 | 67.79 | 74.62 | 78.53 | 81.42 | 43.60 | 53.70 | 62.00 | 65.55 | 68.19 | 43.46 | 52.12 | 59.40 | 62.80 | 66.22 |
| DNA-Sim | 72.02 | 78.63 | 84.55 | 86.99 | 88.93 | 48.23 | 57.90 | 64.80 | 67.94 | 70.23 | 44.97 | 52.35 | 60.35 | 64.63 | 68.18 |
| DNABERT-S | 71.36 | 77.93 | 83.37 | 85.81 | 87.77 | 50.25 | 59.41 | 66.07 | 68.92 | 70.75 | 47.83 | 55.83 | 63.01 | 67.12 | 69.82 |

| Model | Synthetic 1 | | | | | Marine 1 | | | | | Plant 1 | | | | |
|---|---|---|---|---|---|---|---|---|---|---|---|---|---|---|---|
| | 1 | 2 | 5 | 10 | 20 | 1 | 2 | 5 | 10 | 20 | 1 | 2 | 5 | 10 | 20 |
| TNF | 43.16 | 54.76 | 68.15 | 74.75 | 78.82 | 26.42 | 38.30 | 51.65 | 57.82 | 60.94 | 24.21 | 33.42 | 44.14 | 50.05 | 54.74 |
| TNF-K | 37.69 | 48.42 | 61.66 | 68.15 | 72.30 | 24.43 | 35.61 | 47.97 | 54.25 | 57.92 | 23.45 | 31.42 | 41.34 | 47.00 | 51.08 |
| TNF-VAE | 33.87 | 46.13 | 60.72 | 68.70 | 74.20 | 22.93 | 33.27 | 45.89 | 52.89 | 57.41 | 20.43 | 29.19 | 39.88 | 46.91 | 52.02 |
| DNA2Vec | 35.00 | 46.13 | 61.20 | 69.44 | 74.55 | 24.17 | 33.63 | 47.06 | 54.29 | 58.45 | 24.03 | 31.81 | 41.93 | 48.29 | 52.97 |
| HyenaDNA | 29.56 | 40.52 | 55.49 | 64.09 | 70.10 | 23.27 | 33.17 | 47.45 | 55.70 | 61.42 | 29.93 | 39.06 | 49.09 | 55.59 | 60.73 |
| DNABERT-2 | 23.87 | 34.01 | 48.29 | 57.24 | 64.39 | 19.06 | 27.41 | 40.19 | 48.69 | 55.52 | 22.02 | 29.70 | 40.11 | 47.59 | 53.34 |
| DNA-Dropout | 20.42 | 27.69 | 39.47 | 46.76 | 53.15 | 14.82 | 21.02 | 30.54 | 37.88 | 44.06 | 19.42 | 25.55 | 34.35 | 40.38 | 45.58 |
| DNA-Double | 33.82 | 44.25 | 57.85 | 66.31 | 72.54 | 26.41 | 36.51 | 49.72 | 57.24 | 62.80 | 23.93 | 33.28 | 45.28 | 53.16 | 59.45 |
| DNA-Mutate | 21.09 | 29.04 | 41.47 | 49.50 | 56.18 | 14.95 | 21.24 | 31.54 | 39.21 | 45.69 | 18.65 | 25.02 | 34.22 | 40.38 | 45.58 |
| Hyena-Sim | 57.58 | 66.16 | 74.08 | 77.74 | 80.49 | 42.86 | 53.40 | 61.51 | 65.27 | 67.65 | 46.35 | 54.27 | 60.39 | 63.79 | 66.66 |
| DNA-Sim | 70.54 | 77.96 | 83.78 | 86.41 | 88.38 | 47.71 | 56.62 | 63.68 | 67.09 | 69.35 | 46.85 | 54.57 | 61.50 | 65.19 | 68.63 |
| DNABERT-S | 69.30 | 77.13 | 82.88 | 85.35 | 87.06 | 49.42 | 58.12 | 64.94 | 67.85 | 69.95 | 49.82 | 57.62 | 64.14 | 67.24 | 69.81 |

| Model | Marine 2 | | | | | Marine 3 | | | | | Marine 4 | | | | |
|---|---|---|---|---|---|---|---|---|---|---|---|---|---|---|---|
| | 1 | 2 | 5 | 10 | 20 | 1 | 2 | 5 | 10 | 20 | 1 | 2 | 5 | 10 | 20 |
| TNF | 27.63 | 39.37 | 51.84 | 57.51 | 60.41 | 22.11 | 31.74 | 43.71 | 50.26 | 54.31 | 24.28 | 35.58 | 48.86 | 54.81 | 58.49 |
| TNF-K | 25.97 | 36.80 | 48.90 | 54.59 | 57.82 | 20.71 | 29.26 | 40.49 | 46.72 | 50.73 | 23.03 | 33.31 | 45.48 | 51.36 | 55.06 |
| TNF-VAE | 24.10 | 34.20 | 46.27 | 52.85 | 56.98 | 19.23 | 27.58 | 38.88 | 45.90 | 50.74 | 21.20 | 31.16 | 43.00 | 49.83 | 54.51 |
| DNA2Vec | 24.37 | 33.77 | 46.83 | 54.06 | 58.12 | 20.41 | 28.16 | 40.03 | 47.09 | 51.77 | 22.43 | 31.17 | 44.49 | 51.45 | 56.07 |
| HyenaDNA | 23.09 | 32.63 | 46.30 | 53.86 | 59.36 | 19.88 | 28.11 | 40.94 | 48.68 | 54.70 | 20.72 | 30.42 | 44.16 | 52.09 | 58.20 |
| DNABERT-2 | 18.11 | 26.57 | 39.05 | 47.18 | 53.46 | 16.04 | 22.95 | 34.17 | 42.03 | 48.92 | 17.23 | 25.27 | 37.72 | 45.97 | 52.73 |
| DNA-Dropout | 14.93 | 21.06 | 30.01 | 37.00 | 42.79 | 12.55 | 17.61 | 25.63 | 31.77 | 37.30 | 13.40 | 18.94 | 28.23 | 34.69 | 40.79 |
| DNA-Double | 25.81 | 35.94 | 48.36 | 55.96 | 61.02 | 22.47 | 30.78 | 43.11 | 50.84 | 56.48 | 24.06 | 34.41 | 47.32 | 54.76 | 60.24 |
| DNA-Mutate | 14.88 | 20.85 | 30.70 | 38.07 | 44.09 | 12.57 | 17.50 | 26.18 | 32.84 | 38.90 | 13.97 | 19.56 | 29.13 | 36.23 | 42.60 |
| Hyena-Sim | 42.75 | 52.24 | 59.92 | 63.25 | 65.65 | 37.91 | 46.51 | 54.87 | 58.80 | 61.40 | 40.18 | 49.99 | 58.34 | 61.98 | 64.83 |
| DNA-Sim | 48.29 | 56.24 | 62.81 | 65.75 | 67.84 | 41.18 | 49.82 | 57.59 | 61.19 | 63.63 | 43.72 | 53.90 | 60.89 | 64.40 | 66.63 |
| DNABERT-S | 50.01 | 58.01 | 64.06 | 66.49 | 68.46 | 42.77 | 51.23 | 58.68 | 61.93 | 64.12 | 45.50 | 55.12 | 62.13 | 65.36 | 67.51 |

| Model | Plant 2 | | | | | Plant 3 | | | | | Plant 4 | | | | |
|---|---|---|---|---|---|---|---|---|---|---|---|---|---|---|---|
| | 1 | 2 | 5 | 10 | 20 | 1 | 2 | 5 | 10 | 20 | 1 | 2 | 5 | 10 | 20 |
| TNF | 23.83 | 32.52 | 42.16 | 48.19 | 52.96 | 20.85 | 30.49 | 41.50 | 49.03 | 54.45 | 21.18 | 30.01 | 41.71 | 49.21 | 54.43 |
| TNF-K | 23.36 | 30.48 | 39.10 | 44.83 | 49.13 | 20.31 | 28.90 | 39.32 | 45.85 | 50.69 | 20.49 | 28.59 | 39.21 | 46.13 | 50.61 |
| TNF-VAE | 19.86 | 28.90 | 39.24 | 45.67 | 51.14 | 18.42 | 26.50 | 37.51 | 44.62 | 51.14 | 18.46 | 26.85 | 38.17 | 45.50 | 51.26 |
| DNA2Vec | 24.10 | 31.57 | 41.04 | 47.42 | 52.14 | 21.49 | 29.63 | 40.20 | 47.19 | 52.53 | 21.92 | 29.43 | 40.22 | 47.70 | 52.56 |
| HyenaDNA | 30.44 | 39.57 | 48.93 | 55.71 | 60.47 | 26.22 | 34.83 | 46.07 | 52.96 | 59.12 | 26.23 | 34.46 | 46.17 | 53.64 | 59.10 |
| DNABERT-2 | 22.36 | 29.75 | 40.32 | 47.50 | 53.33 | 18.75 | 25.90 | 36.95 | 44.80 | 51.66 | 19.42 | 25.95 | 36.98 | 44.96 | 51.33 |
| DNA-Dropout | 19.35 | 26.29 | 34.47 | 40.20 | 45.38 | 16.50 | 22.20 | 30.57 | 36.74 | 42.07 | 16.55 | 22.62 | 30.52 | 37.32 | 42.62 |
| DNA-Double | 24.72 | 34.16 | 45.32 | 53.45 | 59.63 | 22.24 | 31.03 | 43.23 | 51.30 | 58.74 | 22.35 | 30.85 | 42.68 | 51.68 | 58.47 |
| DNA-Mutate | 18.76 | 25.74 | 34.48 | 41.57 | 46.82 | 15.94 | 21.69 | 30.22 | 36.80 | 42.93 | 15.78 | 21.58 | 30.78 | 37.57 | 43.55 |
| Hyena-Sim | 45.45 | 52.78 | 60.04 | 63.88 | 67.19 | 41.27 | 50.02 | 58.21 | 62.80 | 66.03 | 42.14 | 50.28 | 58.68 | 63.51 | 66.41 |
| DNA-Sim | 45.84 | 52.76 | 61.10 | 65.40 | 68.90 | 45.10 | 54.04 | 62.46 | 66.80 | 70.46 | 44.91 | 53.08 | 61.72 | 66.24 | 69.28 |
| DNABERT-S | 49.02 | 56.28 | 63.85 | 67.83 | 70.55 | 47.85 | 56.90 | 64.60 | 68.52 | 71.60 | 48.03 | 56.11 | 64.42 | 68.25 | 70.98 |

