# OpenReview forum: "DNABERT-S: Pioneering Species Differentiation with Species-Aware DNA Embeddings"
_ICLR.cc/2025/Conference — Submitted to ICLR 2025_

### Official Review · Reviewer_n3kh · 2024-10-30

**Soundness:** 3
**Presentation:** 3
**Contribution:** 3
**Rating:** 6
**Confidence:** 4

**Summary:**

This paper introduces DNABERT-S, a modified version of its precursor, DNABERT-2, that is applied to the task of differentiating DNA sequences between different species.

**Strengths:**

- The authors perform comprehensive testing across a variety of scenarios, as well as a thorough set of ablations.
- While each of the separate components - Manifold Mixup, weighted SimCLR, curriculum learning -  have been introduced and utilized previously, this paper appears to be a new application of these methods to genome sequence data.
- The model strongly outperforms baselines on metagenomics binning, species clustering, and species classification.

**Weaknesses:**

- In general, there is a lack of meta-level detail regarding the models that are being compared. It would be helpful to include tables that compare the number of parameters and embedding dimensions for each model/technique used.
- The use of Curriculum Contrastive Learning (C$^2$LR) strategy with the Manifold Instance Mixup (MI-Mix) loss could be an impactful contribution, however not enough work is done to show the utility of these approaches, and whether they are even helpful enough in light of computational tradeoffs. Indeed, removing C$^2$LR appears to change the performance by a small amount (only a -1.13 to -1.17 decrease).
- The paper is missing comparisons to similar models that employ contrastive learning for metagenomic binning tasks.
- It is strange that the authors would not use the benchmarking set up in CAMI II to assess their model performance, given the built-in genome binning benchmark and comparison to SOTA tools.

**Questions:**

- While the authors claim to compare DNABERT-S to the strongest existing methods, there appear to be a number of comparable approaches that they overlook, especially in metagenomics binning. Some examples include COMEBin, a SOTA binning method based on contrastive multi-view representational learning (Wang et. al, 2024) and CLMB (Zhang et. al, 2022). How does DNABERT-S compare to these baselines?
- In the classification tasks, what is the performance of each baseline? While it is helpful to highlight the difference between DNABERT-S and the best-baseline, please include the full table of performance on each of the datasets for each of the models tested (in the appendix), as this information is still a helpful contribution.
- How does Manifold Instance Mixup differ from Manifold Mixup, introduced in Verma et a. (2019)? Please clarify this in the paper.
- Dataset complexity, meaning the number of genomes present, and the relative abundances of those genomes, can often influence the performance of a model in metagenomics binning. How does DNABERT-S perform at metagenomics binning when these two factors are changed? For example, what happens when the number of relative abundances of different species are highly imbalanced?
- Given the well established, recent CAMI II challenge for metagenomic binning (that the authors reference in the paper), how does DNABERT-2 compare to the tools benchmarked during this challenge (see Meyer et. al, 2022)?


Meyer, F., Fritz, A., Deng, Z. L., Koslicki, D., Lesker, T. R., Gurevich, A., ... & McHardy, A. C. (2022). Critical assessment of metagenome interpretation: the second round of challenges. Nature methods, 19(4), 429-440.

Wang, Z., You, R., Han, H., Liu, W., Sun, F., & Zhu, S. (2024). Effective binning of metagenomic contigs using contrastive multi-view representation learning. Nature Communications, 15(1), 585.

Zhang, P., Jiang, Z., Wang, Y., & Li, Y. (2022). CLMB: Deep contrastive learning for robust metagenomic binning. In International Conference on Research in Computational Molecular Biology (pp. 326-348). Cham: Springer International Publishing.

---

> ### Author Response · Authors · 2024-11-18
> **Authors' Reply (Part 1)**
>
> Thank you very much for your insightful review and suggestions!
>
> ## W1: Number of parameters and embedding dimensions for each models/techniques used in comparision
>
>
>
> Thanks for pointing this out. We acknowledge that the number of parameters and embedding dimensions are important parts in the comparsion.
>
> We have included a table in the latest revised version that compares the number of parameters (million), embedding dimension, inference time (seconds), and inference memory (MB) in Appendix E. The symbol "-" denotes that the inference time or memory is not comparable with the LLM-based models.
>
> | Model | Num. Params (M) | Embedding Dim. | Inference Time (Sec.) | Inference Memory (MB) |
> |----------|:----------:|:----------:|:----------:|:----------:|
> |  TNF  |  0  |  256  |  —  |  —  |
> | TNF-K  |  0.026  |  103  |  —  |  —  |
> | TNF-VAE   |  3  |  768  | —  |  —  |
> | DNA2Vec   |  0.026  |  100  | —  |  —  |
> | HyenaDNA |  28.2  |  256  | 11.16  |  995  |
> | NT-v2  |  97.9  |  512  | 19.16  |  1273  |
> | DNABERT-2   |  117  |  768  | 14.27  |  3991  |
> | DNA-Dropout   | 117  |  768  | 14.27  |  3991  |
> | DNA-Double   | 117  |  768  | 14.27  |  3991  |
> | DNA-Mutate  | 117  |  768  | 14.27  |  3991  |
> | DNABERT-S   |  117  |  768  | 14.27  |  3991  |
>
> ## W2: Utility of $\text{C}^{2}$LR and MI-Mix & Computational tradeoffs
>
> Thanks for the suggestions.
>
>
> (i) There is no extra computational cost when incorporating C$^2$LR since the computation costs of MI-Mix and Weighted SimCLR are nearly the same. For DNABERT-S, we train it with SimCLR for 1 epoch and MI-Mix for 2 epochs. For variants without curriculum learning, we train them for 3 epochs with the same loss function.
>
> (ii) Intuition for our proposed curriculum contrastive learning ($\text{C}^{2}$LR) and MI-Mix.
> > Weighted SimCLR teaches the model whether two sequences are similar or not (a binary outcome of 0 or 1), whereas MI-Mix teaches the model how similar two sequences are, assigning a continuous value between 0 and 1.
>
> > MI-Mix method is especially suitable for species differentiation because species similarities are more nuanced than a binary classification. For example, humans are more similar to monkeys than to viruses. We aim for the DNABERT-S embeddings not only to segregate different species but also to reflect the relative similarities among them (placing humans closer to monkeys and further from viruses). Therefore, the regression-like nature of MI-Mix method makes it ideal for our problem.
>
> > However, predicting these finer-grained similarities is challenging, which is why we introduce **Curriculum Contrastive Learning**. We view Weighted SimCLR as a warm-up phase in model training, where the model first learns to segregate different species, and then we fine-tune it to adjust the distances in a more fine-grained manner.
>
> (iii) Further cases study on the benefit of MI-Mix in genomics data.
>
>
> > Based on our motivation behind the method design, we conduct a case study to empirically validate our intuition. Specifically, we collect 50 5000-bp genome sequences from 3 species: human, monkey, and a randomly selected bacteria named Salmonella enterica. We compute the embedding of each genome sequence, and achieve the species embedding by averaging the embedding of all its 50 sequences. We then compute the cosine distance between human-monkey (H-M), human-bacteria (H-B), and monkey-bacteria (M-B). We then compute the relative distance between humans and bacteria (H-B/H-M) and monkeys and bacteria (M-B/H-M). As shown in the table below, models trained with MI-Mix loss naturally segregate very dissimilar species like humans and bacteria further while keeping similar species like humans and monkeys closer. We observe the same pattern in several different bacteria. These case studies can be an illustration of why MI-Mix is more suitable than SimCLR for metagenomics data, besides the scores in the ablation study. We will conduct more comprehensive case studies and discuss them in the revised version. Thanks for this suggestion.
>
>
>
>
> |                                | H-M  | H-B   | M-B   | H-B/H-M            | M-B/H-M             |
> | ------------------------------ | :----: | :-----: | :-----: | :------------------: | :-------------------: |
> | W. SimCLR only                 | $0.0929$ | $0.7310$ | $0.7722$ | $7.87$               |   $8.31$                |
> | MI-Mix only                    | $0.0807$ | $0.8308$ | $0.8907$ | $\bf{10.29}$         | $\bf{11.04}$           |
> | DNABERT-S (W. SimCLR + MI-Mix) | $0.0761$ | $0.7376$ | $0.7649$  | $\underline{9.70}$ | $\underline{10.06}$ |

---

> ### Author Response · Authors · 2024-11-18
> **Authors' Reply (Part 2)**
>
> ## W3 & Q1: Missing comparison with similar models like COMEBin
>
>
> We appreciate this concern and would like to clarify that DNABERT-S serves a distinct purpose from complete metagenomics binning methods, making direct comparisons inappropriate.
> Modern metagenomics binning methods (e.g., MetaBat2 [Kang19] and COMEBin [Wang24]) typically follow a five-step pipeline utilizing three data types:
>
> 1. Generate DNA embeddings from sequences (**contigs**)
> 2. Extract abundance information (**alignment files**)
> 3. Combine DNA embeddings and abundance information to generate final embeddings
> 4. Perform clustering
> 5. Refine results using external features (**e.g., length, single-copy gene markers**)
>
> DNABERT-S specifically targets step 1, rather than end-to-end metagenomics binning. To evaluate its effectiveness, we implemented step 4 using the straightforward approach detailed in Algorithm 1, comparing DNABERT-S against other methods for step 1 only. We deliberately omitted steps 2, 3, and 5 to isolate and assess the impact of DNA embeddings alone.
>
> Consequently, compared to complete binning methods like COMEBin, our approach uses only one-third of the available information (sequences only, without alignments or external features) and employs a simplified clustering algorithm. DNABERT-S is designed to enhance existing binning methods by replacing their step 1 (TNF), which explains our focus on TNF as the primary baseline. As discussed in our response to W4, despite using significantly less information and a simpler clustering approach, we achieve comparable binning performance to SOTA methods on CAMI2, demonstrating DNABERT-S's effectiveness.
>
>
>
> [Kang19] MetaBAT 2: an adaptive binning algorithm for robust and efficient genome reconstruction from metagenome assemblies, PeerJ 7, 2019
>
> [Wang24] Effective binning of metagenomic contigs using contrastive multi-view representation learning, Nature Communications, 2024
>
>
> ## W4 & Q5: Compare with tools directly on CAMI2 benchmark
>
>
> We appreciate this suggestion. As explained above, our implementation utilizes only a subset of available information and omits three steps from the standard pipeline, making direct comparisons with existing binners not entirely appropriate.
>
> Nevertheless, our results are competitive with existing methods. We have compared our approach against official results from SOTA models on the CAMI2 benchmark, including a baseline implementation using TNF (the default DNA embedding method in most metagenomics binners).
>
>
>
> Results for the 'Plant 5' and 'Plant 6' datasets.
>
> | Plant-Associated         | Completeness | Purity   | F1       |
> | ------------------------ | :------------: | :--------: | :--------: |
> | MetaBat 2                | 14.3         | 89       | 24.6     |
> | MetaBinner               | 15.8         | 66.8     | 25.6     |
> | CONCOCT                  | 16.2         | 69.3     | 26.3     |
> | Vamb                     | 0.1          | 100      | 0.2      |
> | MaxBin                   | 20.5         | 81.3     | 32.8     |
> | **TNF(Plant-5)**        | **17.6**     | **33.5** | **16.4** |
> | **DNABERT-S (Plant-5)** | **27.1**     | **55.5** | **33.3** |
> | **TNF(Plant-6)**        | **16.2**     | **29.8** | **17.1** |
> | **DNABERT-S (Plant-6)** | **22.3**     | **46.6** | **30.6** |
>
>
>
>
>
> Results for the 'Marine 5' and 'Marine 6' datasets.
>
> | Marine                   | Completeness | Purity   | F1       |
> | ------------------------ | :------------: | :--------: | :--------: |
> | MetaBat 2                | 19           | 87.9     | 31.2     |
> | MetaBinner               | 23           | 69.4     | 34.5     |
> | CONCOCT                  | 24.7         | 80       | 37.8     |
> | Vamb                     | 0.8          | 99.9     | 1.5      |
> | MaxBin                   | 20.6         | 68.6     | 31.7     |
> | **TNF (Marine-5)**       | **17.9**     | **42.4** | **21.0** |
> | **DNABERT-S (Marine-5)** | **22.8**     | **51.8** | **28.9** |
> | **TNF (Marine-6)**       | **17.9**     | **41.9** | **21.0** |
> | **DNABERT-S (Marine-6)** | **21.7**     | **50.6** | **27.4** |
>
> Our comparisons yield three key insights:
>
> 1. DNABERT-S largely outperforms TNF, demonstrating its superiority as a DNA embedding method
> 2. The performance gap between TNF and complete binning solutions highlights the importance of integrating additional features (e.g., abundance information)
> 3. Despite our simplified implementation using limited information, we achieve comparable performance to comprehensive binning methods
>
> While integrating DNABERT-S embeddings into existing metagenomics binners (replacing TNF) would require substantial engineering effort, our results suggest this would be a promising direction for future work.

---

> ### Author Response · Authors · 2024-11-18
> **Authors' Reply (Part 3)**
>
> ## Q2: Performance of other baselines in classification.
>
> Thanks for asking this. We agree it is also beneficial to include the results of other models in the appendix. We have organized the data and integrated the statistics in Table 14 (`page 23`) in the revised version. Please see below for the statistics on the first datasets of synthetic, marine, and plant-associated datasets.
> | Plant-0             | 1     | 2     | 5     | 10    | 20    |
> | ------------------- | :-----: | :-----: | :-----: | :-----: | :-----: |
> | TNF                 | 24.01 | 32.69 | 43.39 | 48.99 | 53.29 |
> | TNF-K               | 22.83 | 30.58 | 40.55 | 45.57 | 49.58 |
> | TNF-VAE             | 20.63 | 28.8  | 39.38 | 45.96 | 51.1  |
> | DNA2Vec             | 23.98 | 31.35 | 41.46 | 47.41 | 51.96 |
> | HyenaDNA            | 28.15 | 36.97 | 48.2  | 55.24 | 60.04 |
> | HyenaDNA w/ simclr  | 43.46 | 52.12 | 59.4  | 62.8  | 66.22 |
> | DNABERT-2           | 21.04 | 28.16 | 38.5  | 45.46 | 51.99 |
> | DNA-Dropout         | 19.05 | 24.78 | 33.12 | 38.99 | 44.3  |
> | DNA-Double          | 24.56 | 33.09 | 45.06 | 52.91 | 59.57 |
> | DNA-Mutate          | 18.16 | 24.4  | 33.58 | 40.23 | 46.09 |
> | DNABERT-2 w/ simclr | 44.97 | 52.35 | 60.35 | 64.63 | 68.18 |
> | DNABERT-S           | 47.83 | 55.83 | 63.01 | 67.12 | 69.82 |
>
>
>
>
>
> | Marine-0         | 1     | 2     | 5     | 10    | 20    |
> | ------------------- | :-----: | :-----: | :-----: | :-----: | :-----: |
> | TNF                 | 27.65 | 38.81 | 52.4  | 58.86 | 62.59 |
> | TNF-K               | 25.97 | 36.47 | 49.15 | 55.44 | 59.26 |
> | TNF-VAE             | 23.72 | 34.02 | 47    | 53.88 | 58.59 |
> | DNA2Vec             | 24.56 | 34.04 | 47.79 | 55.36 | 60.11 |
> | HyenaDNA            | 23.92 | 33.94 | 47.47 | 55.5  | 61.42 |
> | HyenaDNA w/ simclr  | 43.6  | 53.7  | 62    | 65.55 | 68.19 |
> | DNABERT-2           | 19.5  | 28.45 | 40.64 | 48.98 | 55.67 |
> | DNA-Dropout         | 15.42 | 21.47 | 30.99 | 38.05 | 44.06 |
> | DNA-Double          | 26.76 | 36.84 | 49.98 | 57.68 | 63.29 |
> | DNA-Mutate          | 15.7  | 21.74 | 31.86 | 39.32 | 45.95 |
> | DNABERT-2 w/ simclr | 48.23 | 57.9  | 64.8  | 67.94 | 70.23 |
> | DNABERT-S           | 50.25 | 59.41 | 66.07 | 68.92 | 70.75 |
>
>
>
> | Synthetic-0         | 1     | 2     | 5     | 10    | 20    |
> | ------------------- | :-----: | :-----: | :-----: | :-----: | :-----: |
> | TNF                 | 44.07 | 56.11 | 68.69 | 75.34 | 79.54 |
> | TNF-K               | 39.06 | 50.22 | 62.52 | 68.55 | 72.82 |
> | TNF-VAE             | 34.23 | 47.06 | 61.44 | 69.31 | 75.02 |
> | DNA2Vec             | 35.85 | 46.98 | 61.54 | 69.64 | 75.26 |
> | HyenaDNA            | 30.13 | 41.18 | 54.86 | 64.03 | 70.69 |
> | HyenaDNA w/ simclr  | 59.58 | 67.79 | 74.62 | 78.53 | 81.42 |
> | DNABERT-2           | 24.43 | 34.81 | 48.93 | 58.58 | 65.98 |
> | DNA-Dropout         | 21.09 | 29.39 | 40.8  | 48.38 | 54.44 |
> | DNA-Double          | 34.54 | 46.54 | 59.86 | 67.44 | 73.6  |
> | DNA-Mutate          | 21.27 | 29.92 | 41.78 | 50.31 | 57.2  |
> | DNABERT-2 w/ simclr | 72.02 | 78.63 | 84.55 | 86.99 | 88.93 |
> | DNABERT-S           | 71.36 | 77.93 | 83.37 | 85.81 | 87.77 |
>
>
>
> ## Q3: How does Manifold Instance Mixup differ from Manifold Mixup
>
>
> Thanks for your comment. Here are some clarifications.
>
> 1. Manifold Mixup [Verma19] is a regularization method that linearly interpolates between hidden states and labels of different data samples at a randomly selected layer. It improves deep neural network representations by training on these interpolations. This technique smoothens decision boundaries, flattens class-specific representations, and promotes less confident predictions on unseen data.
> 2. Manifold I-Mix adapts Manifold Mixup for contrastive learning by applying it to the anchor set. It assigns continuous values between 0 and 1 to indicate the similarity between sequences (for both positive and negative pairs). This approach is particularly effective for species differentiation, teaching the model nuanced similarities rather than binary classifications. For example, humans are more similar to monkeys than to viruses. We aim for the DNABERT-S embeddings not only to segregate different species but also to reflect the relative similarities among them (placing humans closer to monkeys and further from viruses). Therefore, the regression-like nature of Manifold I-Mix method makes it ideal for our problem.
>
>
> [Verma19] Manifold Mixup: Better Representations by Interpolating Hidden States, ICML 2019

---

> ### Author Response · Authors · 2024-11-18
> **Authors' Reply (Part 4)**
>
> ## Q4: Performance in unbalanced data
>
>
>
> We agree that the relative abundances of those genomes is very important to model's performance. So we choose the raw samples from CAMI2 as our evaluation data, which already contains largely unbalanced data. We use the 0/25/50/75/100 percentile of number of sequences in each species as the data balance statistics and present them in the table below.
>
>
>
> |      | Plant-5 | Plant-6 | Marine-5 | Marine-6 |
> | ---- | :-------: | :-------: | :--------: | :--------: |
> | 0    | 10      | 10      | 10       | 10       |
> | 25   | 66      | 30      | 115      | 114      |
> | 50   | 190     | 116     | 201      | 223      |
> | 75   | 450     | 413     | 345      | 357      |
> | 100  | 4293    | 4599    | 841      | 915      |
>
>
> As the data is already very unbalanced, and the robustness of clustering results largely depends on the clustering algorithm, we validate the model's robustness to data balancing with K-means clustering and consider 10 datasets used in our clustering & classification evaluation.
>
> For each dataset, we kept 100 species and evaluated species clustering with 3 cases.
> - Case 1: Balanced. We keep 100 sequences in each species.
> - Case 2: Less balanced. We keep 100 sequences in the first 10 species, 90 sequences in the next 10 species, 80 sequences in the next 10, ...., 10 sequences in the last 10.
> - Case 3: Very unbalanced. We keep 100 in the first species, 99 in the second species, ... , 1 in the last species
>
>
> We then set K=100 for K-means and use Adjusted Random Index (ARI) as the clustering metrics. We ran each experiment with 5 random seeds and report the mean and std of the runs. As shown in the Table, DNABERT-S demonstrates relatively robust performance as the data goes from purely balanced to largely unbalanced.
>
>
>
>
> |          | Case 1     | Case 2     | Case 3     |
> | -------- | :----------: | :----------: | :----------: |
> | Plant-0  | 52.05±1.02 | 53.44±0.99 | 53.58±1.68 |
> | Plant-1  | 51.49±0.82 | 49.38±1.01 | 49.75±1.15 |
> | Plant-2  | 50.94±0.98 | 53.50±1.98 | 53.13±2.16 |
> | Plant-3  | 55.74±1.04 | 52.29±1.11 | 51.19±1.45 |
> | Plant-4  | 55.21±1.49 | 55.39±1.40 | 55.84±1.04 |
> | Marine-0 | 46.71±0.56 | 39.87±0.45 | 39.57±0.83 |
> | Marine-1 | 44.95±1.92 | 37.77±0.56 | 36.93±0.70 |
> | Marine-2 | 45.73±0.87 | 40.13±0.34 | 39.81±1.13 |
> | Marine-3 | 37.90±1.29 | 30.39±0.93 | 28.91±1.20 |
> | Marine-4 | 47.63±1.31 | 39.28±0.97 | 38.18±1.58 |
>
>
>
>
>
>
> We really appreciate your reviews and suggestions. We hope our reply can solve your concerns. Please don't hesitate to share any other thoughts.

---

### Official Review · Reviewer_U4Wt · 2024-11-04

**Soundness:** 2
**Presentation:** 3
**Contribution:** 3
**Rating:** 6
**Confidence:** 3

**Summary:**

This paper finetunes the DNABERT-2 model to generate species-aware embeddings from genomic sequences.
Current genome foundation models (such as DNABERT-2) are trained on language-modelling training tasks but do not develop discriminative embeddings.
The authors leverage genome species datasets and contrastive methods to learn embeddings that perform better on both unsupervised and supervised downstream tasks in species differentiation.

They develop a training scheme they name C^2LR for Curriculum Contrastive Learning.
In C^2LR the training of the model is in 2 phases:

Phase 1 : First, a weighted version of SimCLR is used to encourage embedding s from the same species to be near each other. Weighted SimCLR is SimCLR but with with higher weights for negative samples closer to the anchor.

Phase 2 :   Next, they introduce and use a contrastive loss called Manifold Instance Mixup. This is a more challenging task where they mix hidden states in a random layer and predict the proportion of the mix at the output.

They create and share an evaluation benchmark and perform extensive evaluations of the resulting embeddings.

**Strengths:**

This is a well written paper with a clear goal and solid choices in methods.

The main novel contribution is the Manifold Instance Mixup method(MI-Mix). They take previous work (i-Mix) which mix inputs in the batch such as images to create examples. They realize that there are no good ways to mix (blend) DNA sequence s at the input and instead apply the i-Mix methodology to the hidden states at a random layer of the network.

They perform extensive evaluations with baselines from VAE and transformer competitors and perform ablation studies.

The embeddings are clearly beneficial in downstream tasks and useful to the community.

They also create and share a benchmark dataset.

**Weaknesses:**

In table 2, it is clear that MI-Mix performs very well on it's own. The paper would be much simpler and just as convincing re. performance if it focused on MI-Mix and dropped the curriculum and weighted SimCLR. Of course, getting the best result for a foundation model is also important.

In Table 1 there should also be at least one baseline for a model that has also gone through some kind of species differentiation training or finetuning. As noted in the text, baseline models are unlearnable or trained on generic language modelling objectives. As far as I can tell only DNABERT-S has had the luxury of using species labels in its training.

There could be more discussion about the original Manifold Mixup method (that i-Mix this method was partly inspired by), and how it relates to the new Manifold Instance Method.

Typos::

line 435 text overlaps with fig 4

**Questions:**

Do any of the other baseline models see the same or similar labels to those used in the contrastive trianing?

How well does MI-Mix perform on other modalities - such as the original i-Mix task?

---

> ### Author Response · Authors · 2024-11-18
> **Authors' Reply (Part 1)**
>
> Thank you very much for your detailed and insightful review!
>
> ## W1: Intuition for Curriculum Contrastive Learning
>
>
> Thank you for pointing this out. MI-Mix indeed performs very well on its own. The reason we choose to use curriculum contrastive learning in this work it that: $\text{C}^{2}$LR does not involve any extra computational costs or engineering efforts, while lead to slightly better performance. As a foundation model for species differentiation, we aim to get the best possible results. Nevertheless, we agree that using MI-Mix alone is good choice for conceptual simplicity in genome representation learning. We will further discuss this in the revised version.
>
> The intuition behind the proposed curriculum contrastive learning ($\text{C}^{2}$LR) and MI-Mix are:
>
> 1. Weighted SimCLR teaches the model whether two sequences are similar or not (a binary outcome of 0 or 1), whereas MI-Mix teaches the model how similar two sequences are, assigning a continuous value between 0 and 1.
> 2. MI-Mix method is especially suitable for species differentiation because species similarities are more nuanced than a binary classification. For example, humans are more similar to monkeys than to viruses. We aim for the DNABERT-S embeddings not only to segregate different species but also to reflect the relative similarities among them (placing humans closer to monkeys and further from viruses). Therefore, the regression-like nature of MI-Mix method makes it ideal for our problem.
> 3. However, predicting these finer-grained similarities is challenging, which is why we introduce **Curriculum Contrastive Learning**. We view Weighted SimCLR as a warm-up phase in model training, where the model first learns to segregate different species, and then we fine-tune it to adjust the distances in a more fine-grained manner.
>
>
> ## W2: Baselines with species differentiation training
>
>
>
> Thanks for indicating this. Yes, only DNABERT-S has gone through species differentiation training among all the models in previous Table 1. We have results of other models with species differentiation training, such as HyeneDNA trained with our proposed data construction and Weighted SimCLR. We have included it in the Table 1 in the revised version to better reflect it.
>
>
> ## W3: Discussion on original mixup
>
>
>
> Thanks for your suggestions.
> 1. Manifold Mixup [Verma19] is a regularization method that linearly interpolates between hidden states and labels of different data samples at a randomly selected layer. It improves deep neural network representations by training on these interpolations. This technique smoothens decision boundaries, flattens class-specific representations, and promotes less confident predictions on unseen data.
> 2. Manifold I-Mix adapts Manifold Mixup for contrastive learning by applying it to the anchor set. It assigns continuous values between 0 and 1 to indicate the similarity between sequences (for both positive and negative pairs). This approach is particularly effective for species differentiation, teaching the model nuanced similarities rather than binary classifications. For example, humans are more similar to monkeys than to viruses. We aim for the DNABERT-S embeddings not only to segregate different species but also to reflect the relative similarities among them (placing humans closer to monkeys and further from viruses). Therefore, the regression-like nature of Manifold I-Mix method makes it ideal for our problem.
>
>
> [Verma19] Manifold Mixup: Better Representations by Interpolating Hidden States, ICML 2019
>
> ## W4: Typo in Figure 4's layout
>
>
>
> Thank you for bringing this to our attention. We apologize for the oversight. We have modified it in the revised version to optimize the layout.
>
>
>
> ## Q1: Any baseline model see the same labels
>
>
>
> Thanks for your question.
>
> 1. In previous Table 1, none of the listed baselines utilize species differentiation labels, as we consider only existing DNA embedding methods for baselines.
> 2. We have undergone species differentiation training, including HyenaDNA with Weighted SimCLR (Sec. 5.6) and various DNABERT-S variants trained with different loss functions (Sec. 5.5). We primarily view these models as ablation studies focusing on the base model and training loss functions. As discussed in W2, we have also included HyenaDNA with Weighted SimCLR as a baseline in Table 1 in the revised version.

---

> ### Author Response · Authors · 2024-11-18
> **Authors' Reply (Part 2)**
>
> ## Q2: Performance of MI-Mix on the original I-Mix tasks
>
>
>
> Thank you for your question. We conduct experiments on the CIFAR-10 dataset in the I-Mix tasks with two separate models: ResNet-18 and ResNet-50.
>
> Following a similar setting as outlined in the I-Mix tasks (Sec. 4 in [Lee23]), we undertake contrastive representation learning on a pretext dataset and assess the quality of representations through supervised classification on a downstream dataset. In all experiments, we train the model for varying epochs for contrastive learning, specifically 50 or 100. For supervised learning, we train the model for 50 epochs across all settings. We employ the N-pair [Sohn16] as the base contrastive learning method. We integrate I-Mix or MI-Mix with N-pair to compare their performances.  The results are presented as follows.
>
> Table 1: Results for ResNet-18 on the CIFAR-10 Dataset
>
> | Training Epochs | N-pair | +I-Mix | +MI-Mix |
> |----------|:----------:|:----------:|:----------:|
> |  50  |  77.18  |  78.30  |   78.30  |
> | 100   |  81.92  |  83.12  |  82.41   |
>
> Table 2: Results for ResNet-50 on the CIFAR-10 Dataset
>
> | Training Epochs | N-pair | +I-Mix | +MI-Mix |
> |----------|:----------:|:----------:|:----------:|
> |  50  |  80.63  |  81.21  |  81.21   |
> | 100   |  85.59  |  85.78  |  85.89   |
>
>
> As shown in the table, I-Mix and MI-Mix achieve very similar performance on images. This is expected as MI-Mix is specifically motivated by the genomics context. I-Mix is less suitable for genomic sequences since sequences from different species may share common segments. If the embedding mixup happens at the beginning of the model, where no contextual information is involved, it becomes very challenging for the model to distinguish the source of a sequence (whether the common segment comes from species A or B). Consequently, the model can become confused when species share common segments. By mixing at an intermediate layer, the common segments incorporate contextual information, allowing for better differentiation of closely related species during model training.
>
> Yet in visions tasks, it is very unlikely that two images will share common segments. Thus, mixing at intermediate layers (MI-Mix) does not have much benefit over mixing at the begining (i-Mix).
>
> [Lee23] i-Mix: A Domain-Agnostic Strategy for Contrastive Representation Learning, ICLR 2023.
> [Sohn16] Improved Deep Metric Learning with Multi-class N-pair Loss Objective, NeurIPS 2016.
>
>
> Thanks a lot for your comments and suggestions. We hope our response can help solve your concerns with this work. Please don't hesitate to share any other thoughts!

---

### Official Review · Reviewer_UzBN · 2024-11-04

**Soundness:** 3
**Presentation:** 3
**Contribution:** 3
**Rating:** 5
**Confidence:** 4

**Summary:**

The paper introduces DNABERT-S, a genome model focused on species-aware DNA embeddings to differentiate and cluster DNA sequences by species effectively. Building upon DNABERT-2, it incorporates two key innovations: Manifold Instance Mixup (MI-Mix) and Curriculum Contrastive Learning (C2LR). MI-Mix mixes hidden representations at random layers to enhance embedding robustness, whereas C2LR gradually presents increasingly challenging training samples to improve model generalization. Experiments across 23 datasets demonstrate DNABERT-S’s effectiveness, especially in species clustering, metagenomics binning, and few-shot classification tasks, showing that it significantly outperforms baselines, including by doubling clustering performance in Adjusted Rand Index (ARI). The model provides a robust and scalable solution to biodiversity studies and microbiome research in label-scarce environments, addressing limitations of previous genome foundation models in species differentiation.

**Strengths:**

1.	Innovative enhancement of embedding representation for species differentiation: The paper improves the embedding representation of DNA sequences by introducing techniques such as DNA-Dropout and DNA-Double, which enable the model to better distinguish DNA sequences of different species. This improvement enhances the robustness of the embedding and the ability to capture the similarity of DNA structures, significantly improving the accuracy of species clustering and classification.
2.	Improving model generalization using contrastive learning: the paper’s Mixing of Streaming Instances (MI-Mix) and Course Contrastive Learning (C2LR) techniques gradually introduce training samples of increasing difficulty during fine-tuning, allowing the model to adapt more efficiently to species-rich macrogenomic data. This approach improves the model’s generalization ability in environments with scarce labels and high data diversity and is suitable for tasks such as macrogenomic binning and species classification.
3.	Practicality for macro-genomic data: The model is particularly suitable for macro-genomic data and biodiversity research. Through targeted fine-tuning and optimization, DNABERT-S significantly improves its performance on tasks such as species clustering and few-sample classification, providing a powerful tool for microbiomics and biodiversity research.

**Weaknesses:**

1.	Limited Novelty of Methodology: The paper employs Manifold Instance Mixup (MI-Mix) and Curriculum Contrastive Learning (C2LR), which are widely recognized in deep learning, limiting the originality of the methodology. The primary innovation lies in adapting these techniques specifically to metagenomic tasks rather than introducing novel technical advancements (see Sections 3.3 and 5.2). To strengthen this aspect, further evidence could clarify why these specific strategies are especially suited to metagenomics, particularly in addressing the shortcomings of traditional Mixup or contrastive learning for the specific challenges within this domain.
2.	Limited Scope of Comparative Experiments: The paper’s experimental validation is confined mainly to select metagenomic datasets, lacking a broader comparison with other current genomic models and widely used bioinformatics tools, such as database search techniques (refer to the experimental setup). Including these common baselines would provide a more comprehensive assessment of DNABERT-S’s effectiveness, highlighting the model’s practical applicability across diverse tasks and data types.
3.	Insufficient Visual Detail in Figures: In Figure 1, the current marker size obscures certain details, making it difficult to interpret the clustering and classification patterns. Adjusting the marker size could improve visibility, enhancing the visualization of data distribution across different methods.
4.	Figure Layout Issues: Figure 4’s layout partially overlaps with the text, which detracts from the paper’s readability and professionalism. Adjusting the figure’s placement could ensure proper spacing and clear separation between text and visuals.
5.	Ablation Study Lacks Detailed Discussion: While the ablation study indicates a substantial improvement when combining W. SimCLR and MI-Mix, the analysis does not sufficiently explore the mechanisms behind this synergy (see Section 5.3). A more detailed discussion, possibly with illustrative examples, would elucidate why the combined approach enhances data representation, providing stronger support for the method’s efficacy.
6.	Unclear Parameter Justification and Redundancy Reduction: The paper references data filtering criteria, such as selecting only species with at least 100 sequences for classification, but lacks a detailed rationale for these choices. Additionally, the redundancy reduction steps are not thoroughly explained, which could influence the results’ transparency and reliability. Supplementing the parameter selection with explicit reasoning in the data preprocessing steps would enhance the methodological rigor.
7.	Potential Data Leakage in Pre-Training: Given that GenBank serves as a substantial training source, there is a potential risk of overlap between the training and testing datasets. The study does not confirm whether this overlap was checked, which raises concerns about possible data leakage (see Section 5.1). Verifying this aspect and addressing any overlapping data would strengthen the reliability of the results.

**Questions:**

1.	The paper lacks detailed information on hyperparameter selection and the tuning process, particularly regarding how these choices impact overall performance. Could the authors provide further details to clarify the influence of these selections on model stability and performance?
2.	What are the specific advantages of DNABERT-S over existing DNA classification models? A more detailed explanation would help elucidate the model’s unique contributions to the field.
3.	Could this method be extended to other biological sequences (e.g., RNA or protein sequences)? If so, what adjustments would be necessary to adapt to these cases?
4.	It would be helpful if the authors could further explain the measures taken to ensure fair comparisons in their experiments, including steps to prevent data leakage and whether model scales were controlled. While the downstream tasks performed well compared to multiple baselines, training/validation/testing based on sequence identity was not conducted, which could pose a risk of data leakage in this setup.
5.	One of the paper’s focuses is on exploring different embedding methods for DNA sequence classification, using a variety of pre-trained models to enhance classification performance. However, a detailed comparison of time and memory consumption across different embedding methods is missing, especially regarding:
(a)	Time and Memory Consumption of Embedding Methods: Could the authors clarify the computational time and memory usage differences between embedding methods during the training and inference stages?
(b)	Resource Analysis of Different Pre-Trained Models: How do time and memory consumption vary across different pre-trained models, and which models are most advantageous for specific tasks? A more detailed analysis could aid in model selection and optimization.

---

> ### Author Response · Authors · 2024-11-18
> **Authors' reply (Part 1)**
>
> Thank you very much for your detailed and insightful review! Your comments and suggestions are very helpful in improving the quality of our manuscript.
>
> ## W1: Novelty
>
>
>
> Thank you for sharing your perspective on this matter. We agree that providing further illustration behind our method design is important.
>
> 1. **SimCLR or I-Mix?** In summary, SimCLR teaches the model whether two sequences are similar or not (a binary outcome of 0 or 1), whereas I-Mix teaches the model how similar two sequences are, assigning a continuous value between 0 and 1. I-Mix methods are especially suitable for species differentiation because species similarities are more nuanced than a binary classification. For example, humans are more similar to monkeys than to bacterias. We aim for the DNABERT-S embeddings not only to segregate different species but also to reflect the relative similarities among them (placing humans closer to monkeys and further from bacterias). Therefore, the regression-like nature of I-Mix method makes it ideal for our problem. However, predicting these finer-grained similarities is challenging, which is why we introduce **Curriculum Contrastive Learning**. We view SimCLR as a warm-up phase in model training, where the model first learns to segregate different species, and then we fine-tune it to adjust the distances in a more fine-grained manner. In our reply to your W5, we provide a case study to empirically validate this.
> 2. **MI-Mix vs. I-Mix.** I-Mix is less suitable for genomic sequences since sequences from different species may share common segments. If the embedding mixup happens at the beginning of the model, where no contextual information is involved, it becomes very challenging for the model to distinguish the source of a sequence (whether the common segment comes from species A or B). Consequently, the model can become confused when species share common segments. By mixing at an intermediate layer, the common segments incorporate contextual information, allowing for better differentiation of closely related species during model training.
>
> Besides the above methodology, our novelty also lies at data construction and benchmark design.
>
> - **Data Construction.** Unlike well-explored areas like NLP and CV, data construction for DNA representation is non-standard with respect to data source, data augmentation, sequence length, and preprocessing. With inappropriate data construction, the trained model is likely to underperform textual features like TNF, as illustrated in Table 1 and Figures 7–9. We demonstrate the viability of learning species-awareness by treating non-overlapping segments of the same species as positive pairs and analyze the effectiveness of different positive pair construction strategies for DNA sequences.
> - **Benchmark Availability.** There is a lack of standard datasets and evaluation strategies for this problem. DNA datasets are diverse in many aspects, such as being balanced or raw, containing seen or unknown species, being data-scarce or abundant, and consisting of reference or long-read sequences. We have therefore compiled and published a benchmark and evaluation pipeline after iterative refinement to address these challenges.
>
>
>
> ## W2: Scope of experiments
>
>
> Thank you for highlighting this issue.
>
>
> We have conducted the experiments, but due to space limitations, we did not include all of them in the main text. We have highlighted them in the revised version (`line 273-282`). Specifically:
>
> 1. We have compared our model with most of the widely-used and state-of-the-art genomics models, including Nucleotide Transformer v2, DNABERT-2, and HyenaDNA (as shown in Table 1, Figures 3 and 4).
> 2. In Appendix C.4, we compare DNABERT-S with MMSeqs2, one of the state-of-the-art database search methods. We show that DNABERT-S achieves slightly better performance than MMSeqs2 with fewer labeled data, indicating the potential of embedding-based methods in taxonomy classification.
> 3. In Appendix C.8, we also compare DNABERT-S with DNABERT-2 on several genome function prediction tasks and show that species-aware training does not significantly improve genome function predictions.
>
>
>
> ## W3: Insufficient visual detail
>
>
> Thank you for your suggestion. Our aim with this figure is to provide a global view showing that DNABERT-S is able to segregate species into separate clusters. We agree that the marker size was not optimal, and we have adjusted the figure accordingly to improve visual clarity and enhance the visualization of data distribution in the revised version.
>
>
>
>
>
> ## W4: Figure layout
>
>
>
>
> Thank you for bringing this to our attention. We apologize for the oversight. We have adjusted the figure's placement in the revised version.

---

> ### Author Response · Authors · 2024-11-18
> **Authors' reply (Part 2)**
>
> ## W5: Ablation study discussion
>
>
> Thanks for highlighting this! Based on our motivation behind the method design (discussed in reply to W1), we conduct a case study to empirically validate our intuition. Specifically, we collect 50 5000-bp genome sequences from 3 species: human, monkey, and a randomly selected bacteria named Salmonella enterica. We compute the embedding of each genome sequence, and achieve the species embedding by averaging the embedding of all its 50 sequences. We then compute the cosine distance between human-monkey (H-M), human-bacteria (H-B), and monkey-bacteria (M-B). We then compute the relative distance between humans and bacteria (H-B/H-M) and monkeys and bacteria (M-B/H-M). As shown in the table below, models trained with MI-Mix loss naturally segregate very dissimilar species like humans and bacteria further while keeping similar species like humans and monkeys closer. We observe the same pattern in several different bacteria. These case studies can illustrate why MI-Mix is more suitable than SimCLR for metagenomics data, besides the scores in the ablation study.
>
>
>
>
> |                                | H-M  | H-B   | M-B   | H-B/H-M            | M-B/H-M             |
> | ------------------------------ | :----: | :-----: | :-----: | :------------------: | :-------------------: |
> | W. SimCLR only                 | $0.0929$ | $0.7310$ | $0.7722$ | $7.87$               |   $8.31$                |
> | MI-Mix only                    | $0.0807$ | $0.8308$ | $0.8907$ | $\bf{10.29}$         | $\bf{11.04}$           |
> | DNABERT-S (W. SimCLR + MI-Mix) | $0.0761$ | $0.7376$ | $0.7649$  | $\underline{9.70}$ | $\underline{10.06}$ |
>
>
>
>
>
> ## W6: Unclear parameter justification and redundancy reduction
>
>
> Thanks for indicating this! We agree that supplementing the parameter selection is helpful and have included them in the revised version (`line 263-272`).
>
> 1. **Datasets for metagenomics binning.** To mimic real-world applications, we use the raw datasets without any data filtering for the metagenomics binning experiments.
> 2. **Datasets for species classification and clustering.** We apply data filtering only to the datasets used for species classification and clustering. Clustering algorithms and few-shot classification are often sensitive to unbalanced data. Therefore, filtering allows us to rule out the data balance factor and fairly compare embedding quality. We chose the threshold of 100 sequences per species based on dataset statistics, which allows us to maintain a sufficient number of species while ensuring enough sequences in each species for reliable analysis. The selection of 100 sequences from each species is purely random, and we will share the code used for this process to enhance transparency.
>
>
>
> ## W7:  Potential data leakage
>
>
> Thank you for expressing this concern. In species differentiation tasks, we consider data leakage to occur when the same species are present in both training and evaluation datasets. We have performed careful validation to prevent this and provided the details in the revised version (Appendix F).
>
> Our experiments use two categories of data: CAMI2 and synthetic datasets. We constructed the synthetic data to ensure they do not include any species present in the training data. For the CAMI2 datasets, due to discrepancies in species annotations between CAMI2 and GenBank, direct validation was challenging. Therefore, we performed an alignment-based estimation using **minimap2**. We aligned each evaluation dataset to the training data and considered sequences with over $90\%$ alignment to the training sequences as present in the training data.
>
> We computed the presence rate of each evaluation dataset as **number of presented sequences / total number of sequences**. It's important to note that different species can share common or highly similar genome sequences, so a non-zero presence rate is expected in real-world scenarios. As a reference, the two synthetic datasets with non-overlapping species have presence rates of $6.88\%$ and $8.92\%$. For the CAMI2 datasets, the plant-associated ones have presence rates between $3.51\%$ and $4.98\%$, which are even lower than the synthetic datasets. The marine datasets have presence rates between $7.99\%$ and $9.45\%$, comparable to the synthetic ones. Based on these statistics, there is negligible species leakage between our training and evaluation data.

---

> ### Author Response · Authors · 2024-11-18
> **Authors' Reply (Part 3)**
>
> ## Q1: Hyperparameter selection
>
>
> Thank you for your question regarding hyperparameter selection.
>
> 1. In our problem, we consider the most important hyperparameters to be sequence length and hidden dimension size, as they are directly related to real-world applications and have a significant impact on performance. In Appendices C.6 and C.7, we present the impact of these two parameters.
> 2. For batch size, we set it to the maximum value possible with BF16 precision on 80GB GPUs, as larger batch sizes generally benefit contrastive learning. Due to the high memory cost of handling long input sequences, we are limited in batch size. Preliminary experiments suggest that our base model, DNABERT-2 [Zhou23], is robust to most hyperparameters, including learning rate, weight decay, and dropout. Therefore, we use the same values suggested in the DNABERT-2 paper.
>
> [Zhou23] DNABERT-2: Efficient Foundation Model and Benchmark For Multi-Species Genomes, ICLR 2023
>
>
>
> ## Q2: Advantage over classification model
>
>
> Thanks for your question. DNABERT-S's most significant advantage over existing DNA classification models is its generalizability to unseen species, which is one of the main motivations behind this work. Traditional classification models are only applicable to the species they are trained on, limiting their utility in real metagenomics research where a large portion of observed sequences belong to unknown species. DNABERT-S, on the other hand, generates embeddings that naturally cluster and segregate sequences from different species, making it applicable to any given DNA sequence regardless of prior knowledge about the species.
>
>
>
>
>
> ## Q3: Extend to other biological sequences
>
>
>
> Thanks for your question. Yes, our method can be readily extended to other biological sequences, such as RNA or protein sequences. The only requirements are:
>
> 1. A deep learning model that generate an embedding for the input biological sequence
> 2. A dataset contains pairs of *similar* sequences. The similarity can be defined in any desired ways (e.g., species, function, and structure).
>
>
>
> ## Q4: Fair comparison
>
>
>
> Thanks for your question. Please refer to our response to W7 for the measures we have taken to prevent data leakage and ensure fair comparisons in our experiments. Regarding model scales, we have tried to reduce their effect by selecting the appropriate version of baseline models with a similar number of parameters. DNABERT-S contains 117 million parameters. Among the deep learning baselines, variants of DNABERT-S (including DNA-Dropout, DNA-Double, DNA-Mutate, and DNABERT-2) have the same number of parameters. For the Nucleotide Transformer, we chose the 97.9 million parameter version to maintain consistency in model size across comparisons.
>
>
>
> ## Q5: Time / Memory / Resources
>
>
>
> Thank you for pointing out the importance of analyzing time and memory consumption. We agree that including this information is helpful, and we have added it to the revised version (Appendix E).
>
> 1. **Time and memory usage of inference.**
>
> Since time and memory usage are primarily impacted by the base model, we compare DNABERT-2, HyenaDNA, and Nucleotide Transformer using sequences of 10,000 base pairs and BF16 precision. The memory usage is measured with a batch size of 1, and time is computed using the largest possible batch size when encoding 512 sequences. As shown below, the models demonstrate similar inference speeds, while DNABERT-2 uses more memory than the Nucleotide Transformer and HyenaDNA.
>
> |             | DNABERT-2 | Nucleotide Transformer | HyenaDNA |
> | ----------- | :---------: | :----------------------: | :--------: |
> | Time (Secs) | 14.27     | 19.16                  | 11.16    |
> | Memory (MB) | 3991      | 1273                   | 995      |
>
>
>
> 2. **Resources for training different models**
>
> Comparing training costs directly is challenging because DNABERT-S is trained upon the pre-trained DNABERT-2, and different genomics models are trained with different datasets. However, for reference, pre-training DNABERT-2 takes approximately 3 days on 8 A100 80GB GPUs, while training DNABERT-S takes around 2 days using the same resources.
>
>
>
> 3. **Model selection**
>
> Based on our empirical study, DNABERT-S is most advantageous for tasks where species differentiation is important. For single-species tasks, such as promoter prediction on the human genome, genome foundation models like DNABERT-2 and Nucleotide Transformer are preferred choices. For tasks involving extra-long sequences (e.g., 1 million base pairs), HyenaDNA offers advantages due to its ability to handle longer sequences efficiently.
>
>
> Thanks again for all the suggestions. We hope our reply can solve your concerns. Please don't hesitate to share any other thoughts!

---

### Author Response · Authors · 2024-11-19
**Global Response (Revision Summary)**

Dear Reviewers,

Thank you for your insightful feedback and the time invested in reviewing our work. We have addressed your inquiries and concerns in our rebuttal and the updated manuscript. The changes in the updated manuscript are **highlighted in blue**.

Following your suggestions, we have enhanced the paper's readability through proofreading and adding more experimental details. For the convenience of reviewers/future readers, this response provides a high-level overview of our contributions.

**Revision Details**

In response to the reviewers' suggestions, we have made several key modifications, summarized as follows:

**Major revisions include:**
1. Included the comparison of the number of parameters, embedding dimensions, inference time and memory for the models in Appendix E. [Reviewers `UzBN` and `n3kh`]
2. Justified the absence of data leakage in Appendix F [Reviewer `UzBN`]
3. Provided detailed classification results for all baselines on all 12 datasets for completeness in Table 14. [Reviewer `n3kh`]
4. Included additional results in Table 1: the results for fine-tuning HyenaDNA and DNABERT-2 using the Weighted SimCLR loss for 3 epochs with the same training dataset used for DNABERT-S. [Reviewer `U4Wt`]

**Minor revisions include:**
1. Adjusted Figure 4's placement. [Reviewers `UzBN` and `U4Wt`]
2. Improved the visual clarity of Figure 1. [Reviewer `UzBN`]
3. Included parameter justification and redundancy reduction (line `263-272`).  [Reviewer `UzBN`]

---

### Meta-Review · Area_Chair_9Kc2 · 2024-12-22

**Metareview:**

The authors introduce techniques like DNA-Dropout and DNA-Double to improve embedding representations, enhancing the model's ability to distinguish between species. The paper utilizes contrastive learning techniques, such as MI-Mix and C2LR, to gradually introduce increasingly difficult training samples, improving the model's generalization ability, especially with limited labeled data. The paper was praised for its clear presentation and comprehensive testing across various scenarios.

However, the reviewers also felt that the techniques employed, like MI-Mix and C2LR, are not entirely novel and are adapted from existing deep learning methods, limiting the originality of the methodological contribution. The experimental validation focuses mainly on metagenomic datasets, lacking a broader comparison with other current genomic models and widely used bioinformatics tools.
The ablation study, while showing improvement with combined methods, lacks a detailed exploration of the mechanisms behind the observed synergy The paper lacked detailed explanations for parameter choices and data preprocessing steps, raising concerns about transparency and potential data leakage from the training data.

For these reasons, overall, the reviewers felt the paper is slightly below the acceptance threshold in its current state.

**Additional Comments On Reviewer Discussion:**

- The authors clarified that while they adapt existing deep learning techniques like MI-Mix and C2LR, their primary innovation lies in applying these methods to the specific challenges of metagenomic tasks and addressing the shortcomings of traditional approaches in this domain.

- They also emphasized their novelty in data construction and benchmark design, highlighting the non-standard nature of data preparation for DNA representation and the lack of standard datasets and evaluation strategies for this problem.

- The authors acknowledged the reviewers' concern and clarified that they conducted experiments with a wider range of models and methods, but did not include all of them in the main text due to space limitations.

 - The authors included additional details in the revised version to clarify their parameter selection and data preprocessing steps.

---

### Decision · Program_Chairs · 2025-01-22

Reject